# Forklift: An Extensible Neural Lifter

**Jordi Armengol-Estapé**[*]
School of Informatics
University of Edinburgh
jordi.armengol.estape@ed.ac.uk

**Rodrigo C. O. Rocha**
Huawei Research

**Jackson Woodruff**
School of Informatics
University of Edinburgh

**Pasquale Minervini**
School of Informatics
University of Edinburgh

**Michael F.P. O'Boyle**
School of Informatics
University of Edinburgh

## Abstract

The escalating demand to migrate legacy software across different Instruction Set Architectures (ISAs) has driven the development of assembly-to-assembly translators to map between their respective assembly languages. However, the development of these tools requires substantial engineering effort. State-of-the-art approaches use *lifting*, a technique where source assembly code is translated to an architecture-independent intermediate representation (IR) — for example, the LLVM IR — and use a pre-existing compiler to recompile the IR to the target ISA. However, the hand-written rules these lifters employ are sensitive to the particular compiler and optimization level used to generate the code and require significant engineering effort to support each new ISA.

We propose Forklift, the first *neural lifter* that learns how to translate assembly to LLVM IR using a token-level encoder-decoder Transformer. We show how to incrementally add support to new ISAs by fine tuning the assembly encoder and freezing the IR decoder, improving the overall accuracy and efficiency. We collect millions of parallel LLVM IR, x86, ARM, and RISC-V programs across compilers and optimization levels to train Forklift and set up an input/output-based accuracy harness. We evaluate Forklift on two challenging benchmark suites and translate 2.5x more x86 programs than a state-of-the-art hand-written lifter and 4.4x more x86 programs than GPT-4 as well as enabling translation from new ISAs.

## 1 Introduction

We are witnessing a rapid diversification of computer hardware to overcome the limits of technology scaling. This diversification has been accompanied by an increase in the number and variety of Instruction Set Architectures (ISAs) (Hennessy & Patterson, 2019). However, for the successful adoption of any new hardware, pre-existing legacy applications must be ported to their new ISAs (Pegoraro, 2023). Unfortunately, legacy software frequently exists only in binary form with no access to the original source code, preventing easy recompilation to new ISAs (Di Cosmo & Zacchiroli, 2017).

Binary translation solves this problem: taking binaries and translating them to new architectures (Rocha et al., 2022; D'Antras et al., 2017). However, developing binary translation tools is a difficult task that requires substantial engineering effort for each new ISA. For each new ISA, the same challenges must repeatedly be overcome, reconstructing information typically destroyed during compilation such as function signatures and pointers, and undoing architecture-specific steps such as register allocation and vectorization (Cifuentes et al., 2002). These tasks are non-trivial to overcome as compilers are not designed to be bidirectional — and must be redone for each new architecture.

---

[*]Corresponding author.

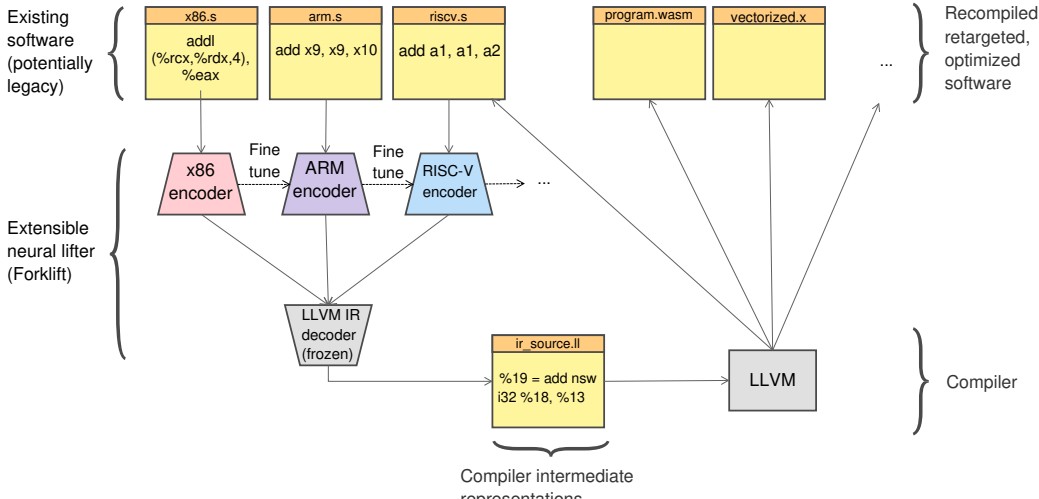

Figure 1: Forklift *lifts* source x86, ARM, RISC-V code into the intermediate representations (IR) of the LLVM compiler. We leverage the LLVM's ability to compile to a wide range of target ISAs. As the target of Forklift is always LLVM IR, we freeze the decoder and fine-tune existing encoders to incrementally add support for new sources, maintaining accuracy on prior sources.

*Lifting* is the standard approach to this problem, where a manually engineered tool lifts the source binary language into a shared compiler intermediate representation (IR) (Cifuentes et al., 2002). In computational linguistics terms, compiler IR can be thought of as a sort of *interlingua* (Alansary, 2014). The existing compiler infrastructure for this IR can then be used to re-optimize and lower the code to a new binary format. While this removes a significant barrier in binary translator design by reusing existing infrastructure, it still leaves the most challenging part of the problem unsolved. Critically, existing lifters require significant engineering efforts. They must overcome the information loss that compiling entails, and they are typically designed not just for a single ISA but also for a compiler and optimization level.

In this paper, we tackle the heart of this engineering problem: how can we automatically learn lifting? We present Forklift, the first *neural lifter*, implemented as a token-level encoder-decoder Transformer. We pose lifting as a sequence-to-sequence task in which the model is directly fed with assembly instructions and is expected to generate the equivalent IR of the LLVM compiler. Unlike previous work in learnable binary translation (Lee et al., 2024), which translates directly to the target assembly rather than lifting, we sidestep the need to learn to generate code for every translation direction. Once Forklift lifts code to LLVM IR, we can leverage the existing power of LLVM to re–optimize and recompile to the desired target, directly benefiting from the LLVM ecosystem.

We collect a million-scale parallel dataset of LLVM IR, x86, ARM, and RISC-V for several compilers and optimization levels to train Forklift and set up an I/O-based accuracy harness.

We evaluate Forklift[1] on two challenging benchmark suites against a strong hand-written lifter, Lasagne (Rocha et al., 2022), and a state-of-the-art Large Language Model (LLM), OpenAI's GPT-4 (OpenAI et al., 2024). Forklift outperforms existing approaches *without requiring manual engineering effort*, automatically learning the intricacies of each ISA, compiler, and optimization level. On optimized x86 code, Forklift is 2.5x more accurate than Lasagne and 4.4x more accurate than GPT-4. Forklift is able to perform on new ISAs in a way traditional tools are not. Furthermore, as we lift all assembly targets to LLVM IR, we freeze the decoder and incrementally fine-tune the encoder for each new architecture. This

---

[1]Code will be made available at https://github.com/jordiae/forklift.

incremental learning enables parameter sharing in the decoder and transfer learning in the encoder, yielding improved accuracy and efficiency.

## 2 Background

**LLVM IR**   The LLVM project (Lattner & Adve, 2004) is an industry-standard compiler infrastructure. It is designed around an intermediate representations language, LLVM IR, which is a generic static single-assignment (SSA) low-level language akin to assembly. Language developers need only translate their programs into LLVM IR to make use of the pre-existing code-generation backends. LLVM supports all ISAs and has a extensive set of optimizations available. LLVM first applies the desired optimization passes to the IR, and then lowers it to the assembly language of choice.

**Neural code translation**   Neural code translation (Chen et al., 2018; Roziere et al., 2020) is the task of learning a neural network model $T$ that takes as input a unit of code (e.g., a function) in a given language $A$ and outputs an equivalent program in the desired target language, $B$:

$$T : A \mapsto B \tag{1}$$

Each instance of $A, B$ is represented as a sequence of tokens. $a = [a_1, \ldots a_n] \in A, B = [b_1, \ldots b_m] \in B$. For the translation between some source $a \in A$ and some target code $b \in B$ to be correct, we require $a$ and $b$ to be semantically equivalent:

$$M[\![a]\!] \equiv M'[\![b]\!] \tag{2}$$

with $M, M'$ being functions denoting the meaning of programs in $A$ and $B$, respectively. While undecidable in general (Winskel, 1993), in practice, there are looser bounds of equivalence such as *observational equivalence* on finite subsets of input/output pairs.

**Incremental learning**   Standard machine learning approaches train models on a given dataset and then remain frozen. However, the model's information can quickly become outdated, or need to be adapted to a new task. A solution is fine tuning the original model with new data. However, this leads to *catastrophic forgetting* (McCloskey & Cohen, 1989), a phenomenon in which we see a drop in the model's performance on the original task or data it was originally trained on. Thus, one is required to either create a copy of the model for each new task, or fine tune with a mix of data including the original dataset. Both options are impractical. In *incremental learning* (van de Ven et al., 2022; Wang et al., 2024) settings, the machine learning system is explicitly designed to incrementally support new tasks, classification tasks, or domains as the system requirements are updated. For example, adding support for new languages to a multilingual machine translation system (Escolano et al., 2019).

## 3 Extensible Neural Lifting

Our goal is to port legacy assembly to new ISAs in a scalable manner. We assume that we have access to a C compiler for all the source ISAs of interest and to an LLVM-based compiler (Clang) for the target ISA, which is the case in practice. We take functions as the basic unit of translation, following previous work on code translation (Lachaux et al., 2020).

We present our method, `Forklift`, in four key steps. First, we describe how to generate machine learning-scale data to train our model (Section 3.1). Second, we discuss the critical design choice of which specific IR to lift to (Section 3.2). Third, we describe the architecture of `Forklift` (Section 3.3). Finally, we propose a method for continually adding new ISAs to `Forklift` (Section 3.4).

### 3.1 Data generation

Our method requires access to a large C dataset, attainable by the crawling of publicly available repositories. We also require the dataset to be at the function-level, and parallel with the corresponding assembly and LLVM IR functions. However, such functions are rarely compilable. They typically need additional context (inclusion of headers, type definitions, constant definitions). Thus, following da Silva et al. (2021)'s methodology, we take a function-level crawling augmented with synthetic dependencies generated with type inference to make them compilable (Armengol-Estapé et al., 2022).

Previous work, however, lacks a parallel dataset with diverse assembly code. We, therefore, augment the dataset in Armengol-Estapé et al. (2022) with the corresponding LLVM IR and RISC-V code. We also add assembly generated by two compilers at three optimization levels O0, O3, Oz. The resulting training dataset features between 2.7 and 3.3M parallel functions depending on the specific target after deduplication.

### 3.2 Target IR

We wish to select a *single target IR* that a) is portable and b) aids learning.

**IR language**  LLVM IR fits the portability requirement. It is designed to capture all operations provided by traditional processors, while working at a higher level in a machine-independent manner (Lattner & Adve, 2004). It has a low semantic gap to hardware but is generic. Portability is provided by the LLVM compiler framework, which allows programs compiled to the LLVM IR to be recompiled to any target with an existing backend, a feature that existing lifters frequently take advantage of (Rocha et al., 2022).

**IR optimization level**  For the same function, LLVM can emit semantically equivalent IR in different forms. With the O0 setting, Clang will emit unoptimized IR, directly translated from the source code; with O3, the compiler will emit IR optimized for execution time, and with Oz, the compiler will emit IR optimized for code size.

We observe that Oz presents several advantages in terms of easing the learning task. Firstly, its more compact nature leads to shorter sequences for the model. Secondly, Oz transformations act as a *normalizer*. However, in terms of portability, we require the selected IR to be recompilable to the desired optimization level. We exploit the fact that LLVM infrastructure provides many transformations and optimizations. This is empirically confirmed in our benchmarks, where we observe that source functions that were automatically vectorized when compiled with O3, are also often vectorized when first lifted targeting Oz and later recompiled with O3. In Appendix G, we provide examples of these reoptimizations.

Thus, we conclude that our lifting target to be LLVM IR *Oz*. In section 5, we will see the empirical confirmation of this choice.

### 3.3 Neural lifting

We pose neural lifting as a sequence-to-sequence task. Given a dataset $\mathcal{D}$ with $N$ pairs $\{(\mathbf{S_i}, \mathbf{L_i}) | i \in \{0..N-1\}\}$, where $\mathbf{S_n}$ is a function assembly code and $\mathbf{L_n}$ is the corresponding compiler IR code that is equivalent to it, we want to train a model parameterized by $\boldsymbol{\theta}$ with maximum likelihood estimation:

$$\boldsymbol{\theta}^* = \arg\max_{\boldsymbol{\theta}} \prod_{i=0}^{N-1} P(\mathbf{L_i}|\mathbf{S_i}; \boldsymbol{\theta}) \tag{3}$$

In practice, we minimize the negative log-likelihood.

**Model**  We opt for an encoder-decoder Transformer (Vaswani et al., 2017a) using the architectural modifications in BART (Lewis et al., 2019). The first model is trained from

scratch, without any pretraining. We do not fine tune from an existing LLM due to the additional computational cost not being justified given the poor performance of code LLMs on low-level code observed in Lee et al. (2024). If an LLM pretrained on LLVM IR was available,[2] one could opt for starting the training from the LLM as the decoder, and a randomly initialized encoder attached to the decoder via randomly initialized cross-attention modules added to the decoder. Our method is orthogonal to LLM improvements.

**Tokenization** For the vocabulary, we train a Unigram tokenizer (Kudo, 2018) of 16k tokens. We split individual digits and apply domain-specific normalization (such as space and end-of-line normalization, unnecessary assembly boilerplate removal).[3] We note that, unlike assembly languages, LLVM IR has a native `struct` type for user-defined types, like C. This means that the model will learn to generate structs from code that does not have structs, which can lead to complications for the model. In Armengol-Estapé et al. (2024), a similar problem was solved with type inference. However, since lifting, unlike decompilation, does not require realistic type names, we instead opt for simplifying the task to the model by normalizing struct names.

**Verifier** We need a method to verify that a given low-level translation is correct. We opt for *observational equivalence*, i.e., unit tests. While unit tests don't formally guarantee semantic equivalence, in practice passing unit tests with good coverage strongly correlates with correctness. Due to the imperative, low-level nature of the code we tackle, we implement a code evaluator that aside from return values, also checks for side effects on arrays, struct pointers, and global variables. We do not check for side effects involving system calls. We assume that the desired functions have reference unit tests, which is an increasingly realistic assumption given the proliferation of automatic unit test generation methods (Kind et al., 2022; Siddiq et al., 2024).

### 3.4 Extensibility

We tackle *incrementally multi-source* neural lifting, that is, we need to incrementally support multiple assembly languages. In this work, we start training an x86 lifter, as x86 is a mainstream ISA. We, then, freeze the decoder, copy the x86 encoder and fine-tune it for ARM. Afterwards, for a new architecture, e.g., RISC-V, we proceed similarly starting from the previous ARM encoder, motivated by the similarities between ARM and RISC-V.

## 4 Experimental framework

To evaluate the effectiveness of our approach, we train neural lifters both from scratch and with the extensibility approach outlined in Section 3.4. We evaluate the models against a state-of-the-art LLM, ChatGPT4 (OpenAI et al., 2024), and a strong handwritten baseline, the lifter used in Lasagne (Rocha et al., 2022) using input/output accuracy.

**Baselines** As our first baseline, we use the lifter in Lasagne (Rocha et al., 2022), which is a state-of-the-art hand-written binary translator built by extending the LLVM-Mctoll lifter (Yadavalli & Smith, 2019). Since this lifter is coupled with a disassembler (similar to Unix's `objdump`), we provide it with an object file. As an LLM baseline, we use the most recent version[4] of GPT-4 (OpenAI et al., 2024). We provide GPT-4 with the textual assembly of the function, and experiment with different prompts.

**Evaluation** We provide each translation system with the source assembly function. We then take the output LLVM IR, compile it back, and call the function from a C interface with

---

[2]After the submission of this article, an LLM pretrained on LLVM IR was released (Cummins et al., 2024). In Section 6, we discuss their results and how they compare to `Forklift`'s.

[3]Unigram was shown to outperform BPE in Bostrom & Durrett (2020), but it is out of scope of this work to perform an ablation study on tokenization strategies.

[4]Most recent version of ChatGPT4 as of March 2024.

| Lifter | x86 | | ARM | | RISC-V | |
|---|---|---|---|---|---|---|
| | Exe | Synth | Exe | Synth | Exe | Synth |
| Lasagne | 30.26% | 37.86% | 1.03% | 0.97% | - | - |
| GPT-4 | 17.07% | 0.95% | 22.74% | 0.97% | 16.11% | 3.37% |
| Forklift | **75.57%** | **51.46%** | **75.31%** | **67.96%** | **71.61%** | **67.42%** |

Table 1: Result summary. The columns represent input-output accuracy for each ISA on each benchmark set for Clang optimized assembly (O3) translation into LLVM IR. Lasagne does not support RISC-V, and has limited support for ARM. On ExeBench in x86, Forklift outperforms Lasagne by 2.5x, and GPT-4 by 4.4x.

the original context of the function. We then check for input/output accuracy using the unit tests from the corresponding evaluation benchmarks, following the verifier implementation described in Section 3. For our models, we sample with beam search and evaluate the top 5 hypotheses.

**Benchmarks** We evaluate our approach on two benchmarks suites: the program synthesis benchmark in Collie et al. (2020), Synth, with a curated set of features in C (e.g., numeric arrays and strings) and a subset of 512 functions in ExeBench (Armengol-Estapé et al., 2022),[5] scrapped from public repositories, to allow wider-scale evaluation and interaction with user-defined types and external function calls. Synth has the interesting characteristic of being rich in automatically vectorizable functions, i.e., loops that the compiler automatically vectorizes when using e.g. the O3 optimization code. While faster, this further complicates recovering the original LLVM IR. For a realistic, yet challenging evaluation, for all ISAs, we evaluate on assembly generated with the O3 optimization level, considerably more complex than unoptimized, O0. To avoid train-evaluation overlaps, the benchmarks in Synth were already filtered out from the train set in Armengol-Estapé et al. (2022) in the original dataset; the same work performed exact C token-level decontamination to minimize train-test overlaps. In this work, we apply additional deduplications using assembly tokens rather than C tokens (unlike Armengol-Estapé et al. (2022)) to search for duplicates, given the fact that different C functions could be lowered to identical assembly.

**Implementation** We implement our models in PyTorch (Paszke et al., 2017) using Hugging-face Transformers (Wolf et al., 2020). We build an encoder-decoder Transformer (Vaswani et al., 2017a) with the architectural modifications in Lewis et al. (2019), for a total of 153M trainable parameters. We use a context window of 2048 tokens for both source and target sequences. For further details please see Appendix A.

## 5 Results, discussion, and analysis

Table 1 shows the overall performance of the lifters across source ISAs and benchmark suite when translating between optimized assembly (O3), generated by the Clang compiler, into LLVM IR. Forklift consistently performs well across the ISAs with a stable accuracy on Exebench of 71 to 75%. Lasagne has at best half this accuracy, achieving this only on x86. The Lasagne ARM port is considerably weaker, demonstrating the difficulty of manual lifters. GPT-4 is not competitive in this task, highlighting the intrinsic difficulty of lifting and hinting at a scarcity of low-level code in mainstream LLMs pretraining data. However, it obtains non-trivial performance on Exebench, ranging between 16% and 22%.

The Synth dataset is especially challenging to lift from optimized assembly, as a considerable amount of the loops are vectorized. Vectorized assembly is typically longer and difficult to get right (custom vector instructions, padding, etc). GPT-4 especially struggles in this benchmark. Instead, Forklift achieves accuracies between 51% and 67.42%. The performance is the lowest on x86, as x86's vectorizations typically require more instructions than their ARM counterparts.

---

[5]The first 512 functions in the test set, to minimize the overhead incurred in the executing the functions and running the different systems.

In the remainder of this section, we analyze the impact of the a) learned LLVM IR target, b) incremental learning, c) the original compiler, and d) prompting strategies for GPT-4. Finally, we perform error analysis and discuss the limitations of our approach. Appendix D provides additional results.

## 5.1 What's the optimal LLVM IR target to learn?

| REPRESENTATION | EXEBENCH | SYNTH |
|---|---|---|
| x86 O3 | $467 \pm 1079$ | $1089 \pm 745$ |
| ARM O3 | $525 \pm 1092$ | $540 \pm 339$ |
| LLVM IR O0 | $1314 \pm 1729$ | $934 \pm 465$ |
| LLVM IR O3 | $956 \pm 1866$ | $2470 \pm 1811$ |
| LLVM IR Oz | $814 \pm 1014$ | $458 \pm 194$ |

Table 2: Mean length and standard deviation (model tokens). Note the considerable variation.

Table 2 shows that while LLVM IR is generally more verbose than x86 and ARM assembly, LLVM IR Oz is the most compact representation among other IRs, and it is even less verbose than ARM and x86 on Synth, due to the presence of vectorizations, which requires much setup code. Table 3 shows the evaluations of neural lifters from x86 using different IR optimization levels, compared to direct translation from x86 to ARM. The IR Oz model is almost on par with the IR O3 one on ExeBench, but it is dramatically better on Synth. This, together with the fact that IR can be recompiled to the ISA of choice, and that vectorized assembly can be regenerated from IR Oz ,[6] makes IR Oz the preferred target for `Forklift`.

| TRANSLATION | EXEBENCH | SYNTH |
|---|---|---|
| x86 O3 $\rightarrow$ ARM O3 | 71.89% | 26.21% |
| x86 O3 $\rightarrow$ LLVM IR O0 | 63.53% | 32.04% |
| x86 O3 $\rightarrow$ LLVM IR O3 | **75.93%** | 8.74% |
| x86 O3 $\rightarrow$ LLVM IR Oz | **75.57%** | **51.46%** |

Table 3: Direct and best lifting target experiments.

## 5.2 How does incremental learning and decoder sharing affect the results of Forklift?

Table 4 shows the performance of `Forklift` (with the extensibility methodology described in Section 3.4 and decoder sharing) across ISAs. As a baseline we compare to models trained from scratch. The incremental approach has the advantage of a smaller parameter count due to the shared decoder. However, we observe additional advantages. Firstly, there is a a faster convergence (see appendix for details), which would help in scenarios with restricted data or compute. Secondly, accuracy slightly improves on x86 ARM and significantly improves on RISC-V, which we attribute to the regularization effect of the fixed decoder and the transfer learning of the encoder.

| Lifter | ARM | | RISC-V | |
|---|---|---|---|---|
| | Exe | Synth | Exe | Synth |
| From scratch | 71.15% | 67.96% | 63.78% | 24.72% |
| `Forklift` | **75.31%** | 67.96% | **71.61%** | **67.42%** |

Table 4: Incremental learning. The columns represent the accuracy for each approach when lifting ISA to LLVM IR Oz across benchmark sets. From scratch trains a new encoder/decoder for each ISA. `Forklift` has a fixed decoder. The encoder for ARM `Forklift` is pretrained with the x86 encoder and RISC-V `Forklift` encoder is pretrained with the prior ARM `Forklift` encoder.

---

[6]See examples in the Appendix G. Appendix B explains the optimization levels.

Table 5 shows a comparison between a full fine-tuning approach vs the frozen decoder approach (`Forklift`). There is no clear winner in terms of accuracy, with one model being slightly better on Synth and the other being slightly better on ExeBench. We prefer the frozen decoder one due to the decoder parameter sharing, leading to lesser overall parameter count.

| MODEL | EXE | SYNTH |
|---|---|---|
| Neural full-FT | 72.37% | **69.90%** |
| Forklift | **75.31%** | 67.96% |

Table 5: Full fine-tuning vs frozen decoder on ARM.

### 5.3 Can we adapt to new compilers?

Hand-written lifters can be sensitive to the original compiler that generated the assembly. Lasagne was originally evaluated with Clang and its performance significantly drops when using GCC-generated code (see Appendix, Table 11). Following a similar approach for extending `Forklift` to ARM and RISC-V with Clang, we fine tune the original x86 encoder with GCC-generated assembly, and outperform Lasagne (Table 6).

| MODEL | EXE | SYNTH |
|---|---|---|
| Lasagne | 22.64% | 35.24% |
| Forklift | **64.97%** | **53.85%** |

Table 6: Compiler adaptation results: Evaluation on x86 03 compiled with GCC.

### 5.4 How does prompting affect GPT-4's performance?

Table 7 shows the performance of GPT-4 with different prompting strategies. First, we simply ask GPT-4 to translate the assembly function into the corresponding LLVM IR. We observe that GPT-4 especially struggles when literally translating architectural intrinsics.Thus, we explicitly ask the model not to use intrinsics and come up with the original LLVM IR code. We also experiment with one-shot learning, which slightly degrades performance. Including prior experiments implies feeding costly long sequences to the model. We use the best-performing prompt, `Prompted no intrinsics` strategy, for the rest of the experiments.

| Prompt | Exe | Synth |
|---|---|---|
| Simple prompt | 12.87% | 0.95% |
| Prompted no intrinsics | **17.07%** | 0.95% |
| Prompted no intrinsics + One-shot | 12.28% | 0.95% |

Table 7: GPT-4 performance on Synth lifting from x86, breakdown.

### 5.5 Error analysis

Table 8 shows the correlations between different features (e.g., compilability of the predicted translation) and the input/output accuracy of each system on x86. We refer to Appendix F for similar tables on other architectures.

We observe that the main feature driving Lasagne failure is vectorization (`vectorized`) on Synth, and global variables (`has_globals`) on ExeBench. The correlation with compilability (`compiles`) of generated code is strong on Synth but weaker on ExeBench. This suggests that on Synth, when Lasagne's LLVM IR compiles, it is likely to be correct. However, on ExeBench, we encounter runtime crashes in many cases, leading to a lack of input/output accuracy even when it compiles. In Appendix E, we provide additional results and analysis for Lasagne.

`Forklift`'s input/output accuracy generally correlates with compilability, which is an interesting feature because it minimizes the unreliability of having compilable yet subtly

incorrect translations. The length of the original C code (`c_length`) is negatively correlated with input-output accuracy, suggesting an increased difficulty with longer sequences.

|  | Synth | | ExeBench | | |
|---|---|---|---|---|---|
|  | Lasagne | Forklift | Lasagne | GPT4 | Forklift |
| compiles | 0.94 | 0.77 | 0.25 | 0.70 | 0.86 |
| edit_sim | 0.29 | 0.50 | 0.26 | 0.46 | 0.43 |
| c_length | -0.14 | -0.31 | -0.10 | -0.23 | -0.44 |
| n_fun_args | -0.36 | -0.09 | 0.22 | -0.21 | -0.18 |
| has_float | -0.10 | -0.04 | 0.08 | -0.02 | -0.08 |
| has_strings | 0.28 | -0.08 | 0.05 | -0.04 | -0.09 |
| vectorized | -0.75 | 0.12 | | | |
| has_globals | | | -0.87 | 0.14 | -0.05 |

Table 8: Pearson correlations between different features and the input/output accuracy of each system on Synth and ExeBench with Clang O3 for x86: `compiles` (does the code generated by the system compile?); `edit_sim`, the normalized edit similarity (i.e., $1 -$ `edit_dist`, with `edit_dist` normalized by length) of the generated code with respect to ground truth LLVM IR; `c_length`, the character length of the original C function; `n_fun_args`, the number of arguments of the function; `has_float`, whether the function has floating point types; `has_strings`, whether the function deals with strings; `vectorized`, whether the function's assembly is vectorized; and `has_globals`, whether the function uses global variables. GPT-4 correlations for Synth are omitted due to its almost 0.0% accuracy on this benchmark. Similarly, the `vectorized` column is omitted from ExeBench due to its sparsity on its test set, and `has_globals` is omitted from Synth as its functions have no global variables.

### 5.6 Limitations

We note the following limitations of our system. Firstly, we tackle functions with a maximum length of 2048 tokens as opposed to full programs with unbounded lengths. Secondly, `Forklift` is fed with textual assembly, for which we assume the presence of a correct disassembler, which is a typical assumption in the literature (Hosseini & Dolan-Gavitt, 2022). Thirdly, `Forklift` does not formally guarantee the semantic equivalence of its translations. Fourth, for the incremental learning strategy we start from the intuition that the encoder is essentially in charge of understanding the source assembly, while the decoder is mainly concerned with IR generation; however, since the system is trained end-to-end, both roles might be more intertwined. Finally, this work was performed under a limited compute budget, and we were not able to pretrain an LLVM IR LLM to work as the decoder.

## 6 Related work

**Binary translators and lifters** A large number of binary translators exist for a wide range of tasks, from optimizing code (Bruening, 2004; Bansal & Aiken, 2008) to instrumenting code (Nethercote & Seward, 2007; Luk et al., 2005). Common approaches are to do this statically (Engelke & Schulz, 2020; Reidel et al., 2021), where binaries are translated ahead of time, and dynamically (Bellard, 2005; Altman & Ebcioglu, 2000), where binaries are translated on the fly. However, these approaches are challenging to port to new architectures (Hazelwood & Klauser, 2006; Chen et al., 2013; Bansal & Aiken, 2008). LLVM IR has been used as a target-independent language for many lifters (Shen et al., 2012). Lasagne (Rocha et al., 2022) introduces a state-of-the-art x86 → LLVM IR lifter by extending the industry-grade LLVM-McToll lifter, and leverages this lifter to build an effective static binary translator x86 → ARM.

**Neural approaches** Transformer-based (Vaswani et al., 2017b) approaches are state-of-the-art in code translation (Lachaux et al., 2020), and LLMs (Brown et al., 2020) have further improved (Chen et al., 2021a). However, even code-specific LLMs still struggle with low-level languages (Lee et al., 2024), as developers don't typically upload assembly to the

Internet. Rather, low-level representations of code are *implicitly* present on codebases. Domain-specific Transformers have been trained in decompilation settings (Chen et al., 2021b; Armengol-Estape et al., 2024; Tan et al., 2024). Unlike decompilation, lifting is more concerned with obtaining accurate and fast portable code rather than readability in a user-level language.

More recently, Cummins et al. (2023) trained an LLM to predict LLVM IR for code size optimization. However, the model cannot handle assembly language and is not publicly available. Guess&Sketch (Lee et al., 2024) tackled RISC-V $\leftrightarrow$ ARM with an encoder-decoder Transformer augmented with a symbolic engine to fix partially correct translations. However, they evaluate on unoptimized assembly and are limited by the quadratic complexity of all translation directions. Our work is orthogonal to theirs and `Forklift` could benefit from the symbolic augmentation, which in our case would involve less engineering work to implement due to only having to support one target language (LLVM IR). Starcoder2 (Lozhkov et al., 2024) is another recent LLM work that includes LLVM IR in the training set, albeit the IR is not paired with the corresponding assembly, and the overall proportion of runtime-optimized assembly is tiny. Relatedly, Ivanov et al. (2024) introduced a large executable LLVM IR dataset based on Grossman et al. (2024)'s compilable dataset.

After the submission of this article, Cummins et al. (2024) introduce pretrained LLMs on LLVM IR and showcase {x86, ARM} lifting as one of their fine-tuning use cases. While the larger scale of their models shows promise (albeit at higher cost), direct comparison with our approach is difficult due to the different granularity (functions vs. files) and metrics used. For example, `Forklift` obtains better compilability (64.08%-80.88%[7]) than the Meta LLM Compiler (49.4%), but in different setups. Additionally, unlike Cummins et al. (2024), `Forklift` evaluates on assembly optimized for execution time and reports input-output accuracy. As future work, it would be interesting to combine both approaches (initializing `Forklift`'s decoder with the weights in Cummins et al. (2024)).

**Incremental learning and interlingua machine translation** Escolano et al. (2020) propose jointly training a multilingual translation system with a per-language encoder and decoder, resulting in a system with shared latent space but no sharing of parameters, inspired by *interlingual* machine translation (Alansary, 2014). Then, they can incrementally add languages by freezing an existing encoder-decoder pair and training the new ones. Unlike natural languages, assembly languages do have an explicit, complete interlingua available, LLVM's intermediate representations. Thus, we can have a single frozen decoder and incrementally add new encoders which in our case we, unlike Escolano et al. (2020), fine tune from existing encoders rather than from scratch. Interestingly, Szafraniec et al. (2023) also exploit the interlingual nature of LLVM IR, but as a pivot language for improving the training of code translation models between source languages.

## 7 Conclusions

This paper introduces `Forklift`, the *first neural lifter*. It effectively translates from optimized assembly into generic compiler intermediate representations. This enables porting legacy applications between ISAs, and reoptimizing them to the new architecture, exploiting existing compiler infrastructure. `Forklift` outperforms Lasagne, a state-of-the-art hand-written lifter, and a GPT-4, a strong general-purpose LLM, and it is capable of efficiently supporting new ISAs. As future work, we suggest replacing `Forklift`'s decoder by an LLM pretrained on LLVM IR, and integrating symbolic approaches.

## Acknowledgments

This work was partly sponsored by Huawei Research UK. We thank the reviewers for their comments and suggestions.

---

[7]As shown in Table 9 in the Appendix D.

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

# A  Implementation details

We use a learning rate of $1 \times 10^{-4}$, with an inverse square root scheduler and $10k$ warmup updates. We train each model with a global batch size of 16 function pairs for a total of $900k$ updates on 4 NVIDIA A100 for a total of 72 hours, using a weight decay of $5 \times 10^{-3}$ as regularization. For the adaptations, we use a learning rate of $5 \times 10^{-5}$ and $5k$ warmup updates, freezing all decoder parameters.[8]

Regarding the training data, we augmented the compilable C dataset in Armengol-Estapé et al. (2022) with the required assembly and LLVM IR targets. Aside from the token-based C deduplication performed in the original dataset, we additionally perform an assembly-based deduplication, which leads to finding further duplicates (different C code can lead to the same assembly), and use a similar strategy to prevent data leakage to the evaluation sets. We also filter for length, but only for the training set. For a realistic evaluation, we do not filter for length on evaluation sets for any of the systems. After filtering, we are left with about 2.7-3.3M parallel functions depending on the translation pair. We use the validation set to monitor training and select the best checkpoint.

As for the tokenizer, we train a Unigram tokenizer with vocabulary of 16k tokens. using the `tokenizers` library. [9] For incrementally adding new languages to the vocabulary, we would train new tokenizers and merge them back into the shared vocabulary, training only the embeddings of the new tokens (the ones without overlap with previous vocabulary). To simplify the implementation, we do a one-off shared vocabulary for LLVM IR, x86, ARM, and RISCV).

There are several variants of ARM. For evaluation, we use the target triple `aarch64-linux-gnu`. However, Lasagne does not have 64 bit support for ARM. Thus, for a fair evaluation, for Lasagne we use the target used in their tests, `arm-linux-gnueabi`.

Before evaluating the translation systems, we check that the original function compiles in our host machine, and discard those which don't for all systems. Depending on the ISA and benchmark, the number of discarded functions ranges between 3% and 23%.

# B  Compiler optimization levels

These are the compiler (Clang, GCC) optimization levels used in this work:

- O0: Unoptimized code.
- O3: Aggressive execution in terms of execution time.
- Oz: Aggressive optimization in terms of code size.

The length of IR target varies across formats and benchmark suite. O0 is expected to be more verbose as it is deliberately unoptimized to aid programmer debugging. O3 targets performance and optimizes unnecessary computation away and is hence normally smaller in size. However, for Synth it is considerably larger due to the presence of extensive vectorization which requires much setup code. Oz aims at code size reduction and hence is the smallest of the IRs.

# C  Convergence

Figures 2 (ExeBench) and 3 (Synth) show the faster convergence of the ARM lifter when fine-tuning from x86 as opposed to starting from scratch.

---

[8]Together with the encoder embedding layer, tied with the decoder embeddings.
[9]https://github.com/huggingface/tokenizers

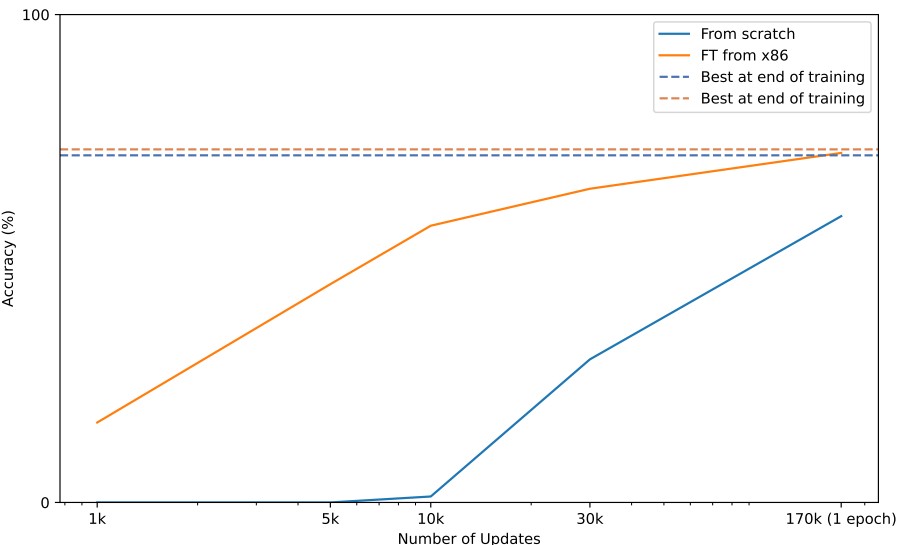

Figure 2: Input/output accuracy on different stages in the training, on ExeBench for ARM.

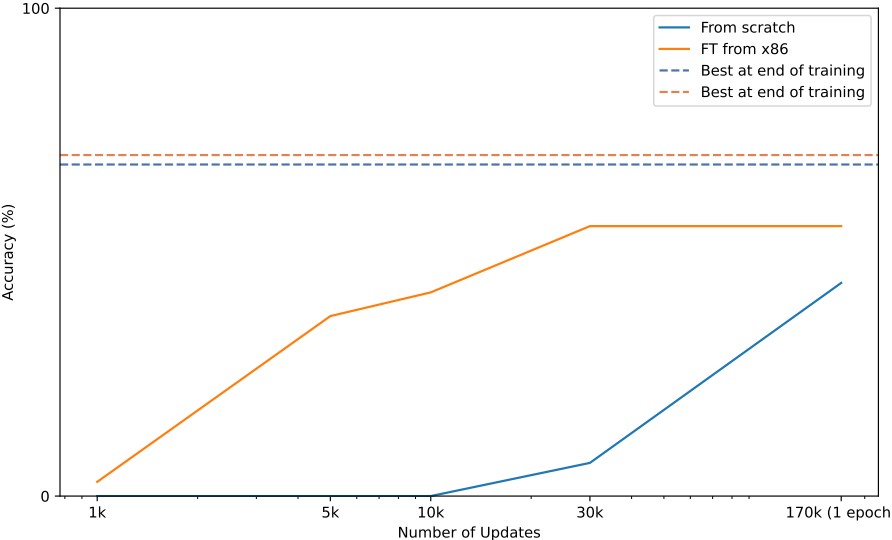

Figure 3: Input/output accuracy on different stages in the training, on Synth for ARM.

# D   Additional results

Table 9 shows the % of compilable translations generated by each system from O3 assembly.

| Lifter | x86 | | ARM | | RISC-V | |
|---|---|---|---|---|---|---|
| | Exe | Synth | Exe | Synth | Exe | Synth |
| Lasagne | 88.89% | 40.78% | 1.03% | 0.97% | - | - |
| GPT-4 | 29.94% | 6.67% | 39.61% | 8.74% | 31.97% | 5.62% |
| Forklift | **80.88%** | **64.08%** | **80.44%** | **78.64%** | **78.26%** | **76.40%** |

Table 9:  Compilability results.

Table 10 shows the normalized edit distance of the generated translations with respect to the ground truth LLVM IR Oz.

| Lifter | x86 | | ARM | | RISC-V | |
|---|---|---|---|---|---|---|
| | Exe | Synth | Exe | Synth | Exe | Synth |
| Lasagne | 0.74 | 0.89 | 1.00 | 1.00 | - | - |
| GPT-4 | 0.73 | 0.62 | 0.67 | 0.65 | 0.68 | 0.68 |
| Forklift | **0.21** | **0.34** | **0.20** | **0.26** | **0.21** | **0.32** |

Table 10:  Edit distance results.

# E Additional Lasagne Results

Given the relatively poor performance of Lasagne, it is worth investigating. Table 11 shows the performance of Lasagne across the experimental parameters. Lasagne performs well in restricted settings, but struggles when moved out of its "comfort zone". On numerical benchmarks where the x86 binary has been generated by Clang with no optimizations, it is able to perform extremely well, outperforming all techniques.

However, its performance drops by two-thirds when changing benchmark suite with more complex data types or compiler optimization level which introduce vector instructions.

The immaturity of the ARM lifter is emphasised across each category. Although it performs best on Clang O0 Synth, the performance is less than half that achieved on x86. Changing compilers again has an overall negative effect, surprisingly though, using GCC with O0 on Exebench is better than using Clang.

| Compiler | Opt | x86 | | ARM | |
|---|---|---|---|---|---|
| | | Exe | Synth | Exe | Synth |
| Clang | O0 | 33.01% | **91.26%** | 10.35% | 39.18% |
| | O3 | 30.26% | 37.86% | 1.03% | 0.97% |
| GCC | O0 | 26.42% | 41.35% | 16.90% | 1% |
| | O3 | 22.64% | 35.24% | 2.11% | 0.97% |

Table 11: Lasagne performance across benchmarks, ISAs, compilers and optimization levels.

# F  Additional analysis

In this section, we include additional input/output accuracy correlation tables. We have excluded the systems for which the accuracies were close to 0 on the corresponding dataset.

| | GPT4 | Forklift |
|---|---|---|
| compiles | 0.64 | 0.85 |
| edit_sim | 0.51 | 0.43 |
| c_length | -0.27 | -0.48 |
| n_fun_args | -0.20 | -0.18 |
| has_float | 0.04 | -0.02 |
| has_strings | -0.02 | -0.10 |
| has_globals | 0.18 | -0.06 |

Table 12: ExeBench feature correlations with input/output accuracy, Clang RISC-V O3

| | Lasagne | GPT4 | Forklift |
|---|---|---|---|
| compiles | 0.89 | 0.67 | 0.87 |
| edit_sim | 0.36 | 0.46 | 0.39 |
| c_length | -0.05 | -0.31 | -0.48 |
| n_fun_args | 0.01 | -0.23 | -0.19 |
| has_float | 0.00 | -0.05 | -0.08 |
| has_strings | -0.03 | 0.03 | -0.16 |
| has_globals | -0.26 | 0.18 | -0.02 |

Table 13: ExeBench feature correlations with input/output accuracy, Clang ARM O3

| | Lasagne | Forklift |
|---|---|---|
| compiles | 0.35 | 0.84 |
| edit_sim | 0.37 | 0.39 |
| c_length | -0.09 | -0.51 |
| n_fun_args | 0.18 | -0.23 |
| has_float | 0.05 | -0.08 |
| has_strings | 0.02 | -0.12 |
| has_globals | -0.77 | -0.07 |

Table 14: ExeBench feature correlations with input/output accuracy, GCC x86 O3

|            | Forklift |
|------------|----------|
| compiles   | 0.80     |
| edit_sim   | 0.31     |
| c_length   | -0.22    |
| n_fun_args | 0.07     |
| has_strings| -0.24    |

Table 15: Synth feature correlations with input/output accuracy, Clang RISC-V O3

|            |        |
|------------|--------|
| compiles   | 0.76   |
| edit_sim   | 0.34   |
| c_length   | -0.37  |
| n_fun_args | -0.18  |
| has_float  | -0.01  |
| has_strings| -0.20  |
| vectorized | 0.21   |

Table 16: Synth feature correlations with input/output accuracy, Clang ARM O3

|            | x86_O3_GCC |          |
|------------|------------|----------|
|            | Lasagne    | Forklift |
| compiles   | 0.98       | 0.74     |
| edit_sim   | 0.55       | 0.21     |
| c_length   | -0.03      | -0.38    |
| n_fun_args | -0.25      | 0.02     |
| has_float  | -0.19      | -0.06    |
| has_strings| 0.17       | -0.13    |
| vectorized | -0.51      | 0.34     |

Table 17: Synth feature correlations with input/output accuracy, GCC x86 O3

# G  Examples

This section provides detailed examples to illustrate where existing lifters fail to accurately lift code and features that `Forklift` struggles to lift. Existing lifters struggle to lift features they were not designed for. For example Lasagne Rocha et al. (2022) fails to lift code that is vectorized, which frequently happens on higher optimization levels. Similarly, we show examples where `Forklift` is able to remove architecture-specific features such as vectorization when it lifts the code, allowing LLVM to re-introduce vectorization for the new targets. Finally, we show an examples where `Forklift` produces incorrect code, and highlight the future work that should be done to improve this.

## G.1  Benchmark `array_inc` (`Forklift` correct, rest incorrect)

This benchmark (Listing 1) increments each element of an integer array by one. When compiled with clang on O3 (Listing 2) and Oz (Listing 3), it is vectorized, as seen by the use of the psubd instructions in the .LBB0_6 block.

`Forklift` lifts this correctly (Listing 4), recreating code in a similar style to Oz (Listing 3). GPT-4 fails to recreate the correct code (Listing 6), producing syntactically correct LLVM IR that does not perform the correct function. Rule-based Lasagne lifter does not support all the vector instructions in the assembly code (Listing 8).

Interestingly, we also provide the O3 recompilations to x86 (Listing 5) and ARM (Listing 6) of the LLVM IR Oz predicted by `Forklift`. Critically, both are vectorized, showcasing that we can recover generic code from an optimized assembly and use LLVM to recompile it with new optimizations.

Listing 1: Original C

```
void array_inc(int *arr, int n) {
  for (int i = 0; i < n; ++i) {
    arr[i] += 1;
  }
}
```

Listing 2: Clang x86 O3

```
        .globl   array_inc              # -- Begin function array_inc
        .p2align        4, 0x90
        .type    array_inc,@function
array_inc:                              # @array_inc
        .cfi_startproc
# %bb.0:
        testl   %esi, %esi
        jle     .LBB0_11
# %bb.1:
        movl    %esi, %eax
        cmpl    $7, %esi
        ja      .LBB0_3
# %bb.2:
        xorl    %ecx, %ecx
        jmp     .LBB0_10
.LBB0_3:
        movl    %eax, %ecx
        andl    $-8, %ecx
        leaq    -8(%rcx), %rsi
        movq    %rsi, %rdx
        shrq    $3, %rdx
        addq    $1, %rdx
        movl    %edx, %r8d
        andl    $1, %r8d
        testq   %rsi, %rsi
        je      .LBB0_4
# %bb.5:
        movq    %r8, %rsi
        subq    %rdx, %rsi
        xorl    %edx, %edx
        pcmpeqd %xmm0, %xmm0
        .p2align        4, 0x90
.LBB0_6:                                # =>This Inner Loop Header: Depth=1
        movdqu  (%rdi,%rdx,4), %xmm1
        movdqu  16(%rdi,%rdx,4), %xmm2
        movdqu  32(%rdi,%rdx,4), %xmm3
        movdqu  48(%rdi,%rdx,4), %xmm4
        psubd   %xmm0, %xmm1
        psubd   %xmm0, %xmm2
        movdqu  %xmm1, (%rdi,%rdx,4)
```

```
        movdqu  %xmm2, 16(%rdi,%rdx,4)
        psubd   %xmm0, %xmm3
        psubd   %xmm0, %xmm4
        movdqu  %xmm3, 32(%rdi,%rdx,4)
        movdqu  %xmm4, 48(%rdi,%rdx,4)
        addq    $16, %rdx
        addq    $2, %rsi
        jne     .LBB0_6
# %bb.7:
        testq   %r8, %r8
        je      .LBB0_9
.LBB0_8:
        movdqu  (%rdi,%rdx,4), %xmm0
        movdqu  16(%rdi,%rdx,4), %xmm1
        pcmpeqd %xmm2, %xmm2
        psubd   %xmm2, %xmm0
        psubd   %xmm2, %xmm1
        movdqu  %xmm0, (%rdi,%rdx,4)
        movdqu  %xmm1, 16(%rdi,%rdx,4)
.LBB0_9:
        cmpq    %rax, %rcx
        je      .LBB0_11
        .p2align        4, 0x90
.LBB0_10:                               # =>This Inner Loop Header: Depth=1
        addl    $1, (%rdi,%rcx,4)
        addq    $1, %rcx
        cmpq    %rcx, %rax
        jne     .LBB0_10
.LBB0_11:
        retq
.LBB0_4:
        xorl    %edx, %edx
        testq   %r8, %r8
        jne     .LBB0_8
        jmp     .LBB0_9
.Lfunc_end0:
        .size   array_inc, .Lfunc_end0-array_inc
        .cfi_endproc
```

Listing 3: Ground truth LLVM IR Oz

```
define dso_local void @array_inc(i32* nocapture %0, i32 %1) local_unnamed_addr #0 {
  %3 = sext i32 %1 to i64
  br label %4

4:                                                ; preds = %8, %2
  %5 = phi i64 [ %12, %8 ], [ 0, %2 ]
  %6 = icmp slt i64 %5, %3
  br i1 %6, label %8, label %7

7:                                                ; preds = %4
  ret void

8:                                                ; preds = %4
  %9 = getelementptr inbounds i32, i32* %0, i64 %5
  %10 = load i32, i32* %9, align 4, !tbaa !2
  %11 = add nsw i32 %10, 1
  store i32 %11, i32* %9, align 4, !tbaa !2
  %12 = add nuw nsw i64 %5, 1
  br label %4
}
```

Listing 4: LLVM IR predicted by Forklift (IO correct)

```
%struct.struct0 = type { i32 }
define dso_local void @array_inc(%struct.struct0* nocapture %0, i32 %1) local_unnamed_addr {
%3 = sext i32 %1 to i64
br label %4
4: ; preds = %8, %2
%5 = phi i64 [ %12, %8 ], [ 0, %2 ]
%6 = icmp slt i64 %5, %3
br i1 %6, label %8, label %7
7: ; preds = %4
ret void
8: ; preds = %4
%9 = getelementptr inbounds %struct.struct0, %struct.struct0* %0, i64 %5, i32 0
%10 = load i32, i32* %9, align 4
%11 = add nsw i32 %10, 1
store i32 %11, i32* %9, align 4
%12 = add nuw nsw i64 %5, 1
br label %4
}
```

Listing 5: Vectorized x86 recompilation of the previous LLVM IR Oz

```
        .globl  array_inc               # -- Begin function array_inc
```

```
        .p2align        4, 0x90
        .type   array_inc,@function
array_inc:                              # @array_inc
# %bb.0:
        testl   %esi, %esi
        jle     .LBB0_11
# %bb.1:                                # %.lr.ph.preheader
        movslq  %esi, %rax
        cmpl    $7, %esi
        ja      .LBB0_3
# %bb.2:
        xorl    %ecx, %ecx
        jmp     .LBB0_10
.LBB0_3:                                # %vector.ph
        movq    %rax, %rcx
        andq    $-8, %rcx
        leaq    -8(%rcx), %rsi
        movq    %rsi, %rdx
        shrq    $3, %rdx
        addq    $1, %rdx
        movl    %edx, %r8d
        andl    $1, %r8d
        testq   %rsi, %rsi
        je      .LBB0_4
# %bb.5:                                # %vector.ph.new
        movq    %r8, %rsi
        subq    %rdx, %rsi
        xorl    %edx, %edx
        pcmpeqd %xmm0, %xmm0
        .p2align        4, 0x90
.LBB0_6:                                # %vector.body
                                        # =>This Inner Loop Header: Depth=1
        movdqu  (%rdi,%rdx,4), %xmm1
        movdqu  16(%rdi,%rdx,4), %xmm2
        movdqu  32(%rdi,%rdx,4), %xmm3
        movdqu  48(%rdi,%rdx,4), %xmm4
        psubd   %xmm0, %xmm1
        psubd   %xmm0, %xmm2
        movdqu  %xmm1, (%rdi,%rdx,4)
        movdqu  %xmm2, 16(%rdi,%rdx,4)
        psubd   %xmm0, %xmm3
        psubd   %xmm0, %xmm4
        movdqu  %xmm3, 32(%rdi,%rdx,4)
        movdqu  %xmm4, 48(%rdi,%rdx,4)
        addq    $16, %rdx
        addq    $2, %rsi
        jne     .LBB0_6
# %bb.7:                                # %middle.block.unr-lcssa
        testq   %r8, %r8
        je      .LBB0_9
.LBB0_8:                                # %vector.body.epil
        movdqu  (%rdi,%rdx,4), %xmm0
        movdqu  16(%rdi,%rdx,4), %xmm1
        pcmpeqd %xmm2, %xmm2
        psubd   %xmm2, %xmm0
        psubd   %xmm2, %xmm1
        movdqu  %xmm0, (%rdi,%rdx,4)
        movdqu  %xmm1, 16(%rdi,%rdx,4)
.LBB0_9:                                # %middle.block
        cmpq    %rax, %rcx
        je      .LBB0_11
        .p2align        4, 0x90
.LBB0_10:                               # %.lr.ph
                                        # =>This Inner Loop Header: Depth=1
        addl    $1, (%rdi,%rcx,4)
        addq    $1, %rcx
        cmpq    %rcx, %rax
        jne     .LBB0_10
.LBB0_11:                               # %._crit_edge
        retq
.LBB0_4:
        xorl    %edx, %edx
        testq   %r8, %r8
        jne     .LBB0_8
        jmp     .LBB0_9
.Lfunc_end0:
        .size   array_inc, .Lfunc_end0-array_inc
```

Listing 6: Vectorized ARM recompilation of the previous LLVM IR Oz

```
        .globl  array_inc               // -- Begin function array_inc
        .p2align        2
        .type   array_inc,@function
array_inc:                              // @array_inc
// %bb.0:
                                        // kill: def $w1 killed $w1 def $x1
        cmp     w1, #1                  // =1
        b.lt    .LBB0_8
// %bb.1:                               // %.lr.ph.preheader
        cmp     w1, #7                  // =7
```

```
        sxtw    x8, w1
        b.hi    .LBB0_3
// %bb.2:
        mov     x9, xzr
        b       .LBB0_6
.LBB0_3:                                // %vector.ph
        and     x9, x8, #0xfffffffffffffff8
        add     x10, x0, #16            // =16
        movi    v0.4s, #1
        mov     x11, x9
.LBB0_4:                                // %vector.body
                                        // =>This Inner Loop Header: Depth=1
        ldp     q1, q2, [x10, #-16]
        subs    x11, x11, #8           // =8
        add     v1.4s, v1.4s, v0.4s
        add     v2.4s, v2.4s, v0.4s
        stp     q1, q2, [x10, #-16]
        add     x10, x10, #32          // =32
        b.ne    .LBB0_4
// %bb.5:                               // %middle.block
        cmp     x9, x8
        b.eq    .LBB0_8
.LBB0_6:                                // %.lr.ph.preheader4
        sub     x8, x8, x9
        add     x9, x0, x9, lsl #2
.LBB0_7:                                // %.lr.ph
                                        // =>This Inner Loop Header: Depth=1
        ldr     w10, [x9]
        subs    x8, x8, #1             // =1
        add     w10, w10, #1           // =1
        str     w10, [x9], #4
        b.ne    .LBB0_7
.LBB0_8:                                // %._crit_edge
        ret
.Lfunc_end0:
        .size   array_inc, .Lfunc_end0-array_inc
```

Listing 7: LLVM IR predicted by GPT-4 in the x86 case (compiles but not correct)

```
@.LC0 = internal constant <4 x i32> <i32 1, i32 1, i32 1, i32 1>

declare void @llvm.memcpy.p0i8.p0i8.i64(i8* nocapture writeonly, i8* nocapture readonly, i64, i1 immarg)

define void @array_inc(i32* %0, i32 %1) {
entry:
  %2 = icmp sle i32 %1, 0
  br i1 %2, label %L1, label %3

3:                                      ; preds = %entry
  %4 = add i32 %1, -1
  %5 = icmp ule i32 %4, 2
  br i1 %5, label %L6, label %6

6:                                      ; preds = %3
  %7 = and i32 %1, -4
  %8 = icmp eq i32 %7, 0
  br i1 %8, label %L9, label %9

9:                                      ; preds = %6
  %10 = zext i32 %7 to i64
  %11 = getelementptr i32, i32* %0, i64 %10
  %12 = add i32 %7, 1
  %13 = icmp sle i32 %12, %1
  br i1 %13, label %14, label %L1

14:                                     ; preds = %9
  %15 = getelementptr i32, i32* %0, i32 %12
  store i32 1, i32* %15, align 4
  %16 = add i32 %7, 2
  %17 = icmp sle i32 %16, %1
  br i1 %17, label %18, label %L1

18:                                     ; preds = %14
  %19 = getelementptr i32, i32* %0, i32 %16
  store i32 1, i32* %19, align 4
  %20 = add i32 %7, 3
  %21 = icmp sle i32 %20, %1
  br i1 %21, label %22, label %L1

22:                                     ; preds = %18
  %23 = getelementptr i32, i32* %0, i32 %20
  store i32 1, i32* %23, align 4
  br label %L1

L6:                                     ; preds = %3
  br label %9

L9:                                     ; preds = %6
  br label %L1

L1:                                     ; preds = %22, %18, %14, %9, %entry, %L9
```

```
  ret void
}
```

Listing 8: Error of Lasagne from x86

```
llvm−mctoll:

/app/llvm−project/llvm/tools/llvm−mctoll/X86/X86AdditionalInstrInfo.h:80:
 mctoll::InstructionKind mctoll::getInstructionKind(unsigned int):

Assertion `Iter != mctoll::X86AddlInstrInfo.end() && "Unknown opcode"' failed
```

### G.2 `str_cspn` (both `Forklift` and `Lasagne` correct)

In this benchmark (Listing 9), the string a is searched for characters that exist in the string s. In this case, we can see that the compiled code is not vectorized, it relies only on simple x86 instructions (Listing 10). As a result, Lasagne is able to work. However, the generated code is verbose: much longer than the original code (Listing 15).

This unnatural code means that during future lowering passes, LLVM has a much more challenging task to generate performant code. We can see that GPT-4 fails to create a compilable sequence of LLVM IR (Listing 16), calling a `strlen` function that is not defined.

Finally, we can see that `Forklift` is able to generate correct lifted code (Listing 12) with one key difference from the original (Listing 11): the injection of a struct type around the string. `Forklift` tends to proactively inject these types due to the GitHub code it was trained on, which is very struct-heavy.

Listing 9: Original C

```
int str_cspn(char *a, char *b) {
  int c = 0;

  for (char *k = a; *k; ++k) {
    for (char *s = b; *s; ++s) {
      if (*k == *s) {
        return c;
      }
    }
      c ++;
  }

  return c;
}
```

Listing 10: Clang x86 O3

```
        .globl  str_cspn                 # −− Begin function str_cspn
        .p2align        4, 0x90
        .type    str_cspn,@function
str_cspn:                                # @str_cspn
        .cfi_startproc
# %bb.0:
        movb    (%rdi), %r9b
        testb   %r9b, %r9b
        je      .LBB0_1
# %bb.2:
        movb    (%rsi), %r8b
        testb   %r8b, %r8b
        je      .LBB0_5
# %bb.3:
        addq    $1, %rsi
        xorl    %eax, %eax
.LBB0_4:                                 # =>This Loop Header: Depth=1
                                         #     Child Loop BB0_8 Depth 2
        movq    %rsi, %rdx
        movl    %r8d, %ecx
        .p2align        4, 0x90
.LBB0_8:                                 #   Parent Loop BB0_4 Depth=1
                                         # =>   This Inner Loop Header: Depth=2
        cmpb    %cl, %r9b
        je      .LBB0_10
# %bb.7:                                 #   in Loop: Header=BB0_8 Depth=2
        movzbl  (%rdx), %ecx
        addq    $1, %rdx
        testb   %cl, %cl
        jne     .LBB0_8
# %bb.9:                                 #   in Loop: Header=BB0_4 Depth=1
```

```
        addl    $1, %eax
        movb    1(%rdi), %r9b
        addq    $1, %rdi
        testb   %r9b, %r9b
        jne     .LBB0_4
        jmp     .LBB0_10
.LBB0_1:
        xorl    %eax, %eax
                                        # kill: def $eax killed $eax killed $rax
        retq
.LBB0_5:
        xorl    %eax, %eax
        .p2align        4, 0x90
.LBB0_6:                                # =>This Inner Loop Header: Depth=1
        cmpb    $0, 1(%rdi,%rax)
        leaq    1(%rax), %rax
        jne     .LBB0_6
.LBB0_10:
                                        # kill: def $eax killed $eax killed $rax
        retq
.Lfunc_end0:
        .size   str_cspn, .Lfunc_end0-str_cspn
        .cfi_endproc
```

**Listing 11: Ground truth LLVM IR Oz**

```
define dso_local i32 @str_cspn(i8* nocapture readonly %0, i8* nocapture readonly %1) local_unnamed_addr #0 {
  br label %3

3:                                                ; preds = %15, %2
  %4 = phi i32 [ 0, %2 ], [ %16, %15 ]
  %5 = phi i8* [ %0, %2 ], [ %17, %15 ]
  %6 = load i8, i8* %5, align 1, !tbaa !2
  %7 = icmp eq i8 %6, 0
  br i1 %7, label %18, label %8

8:                                                ; preds = %3, %12
  %9 = phi i8* [ %14, %12 ], [ %1, %3 ]
  %10 = load i8, i8* %9, align 1, !tbaa !2
  %11 = icmp eq i8 %10, 0
  br i1 %11, label %15, label %12

12:                                               ; preds = %8
  %13 = icmp eq i8 %6, %10
  %14 = getelementptr inbounds i8, i8* %9, i64 1
  br i1 %13, label %18, label %8

15:                                               ; preds = %8
  %16 = add nuw nsw i32 %4, 1
  %17 = getelementptr inbounds i8, i8* %5, i64 1
  br label %3

18:                                               ; preds = %3, %12
  ret i32 %4
}
```

**Listing 12: LLVM IR predicted by Forklift (IO correct, but uses new types)**

```
%struct.struct0 = type { i8 }
define dso_local i64 @str_cspn(%struct.struct0* nocapture readonly %0, i8* nocapture readonly %1)
 local_unnamed_addr {
br label %3
3: ; preds = %17, %2
%4 = phi i64 [ 0, %2 ], [ %18, %17 ]
%5 = phi i32 [ 0, %2 ], [ %19, %17 ]
%6 = getelementptr inbounds %struct.struct0, %struct.struct0* %0, i64 %4
%7 = getelementptr inbounds %struct.struct0, %struct.struct0* %6, i64 0, i32 0
%8 = load i8, i8* %7, align 1
%9 = icmp eq i8 %8, 0
br i1 %9, label %20, label %10
10: ; preds = %14, %3
%11 = phi i8* [ %16, %14 ], [ %1, %3 ]
%12 = load i8, i8* %11, align 1
%13 = icmp eq i8 %12, 0
br i1 %13, label %17, label %14
14: ; preds = %10
%15 = icmp eq i8 %8, %12
%16 = getelementptr inbounds i8, i8* %11, i64 1
br i1 %15, label %20, label %10
17: ; preds = %10
%18 = add i64 %4, 1
%19 = add i32 %5, 1
br label %3
20: ; preds = %14, %3
ret i64 %4
}
```

Listing 13: LLVM IR predicted by Lasagne (IO correct)

```
define dso_local i32 @str_cspn(i64 %arg1, i64 %arg2) {
entry:
  %EAX-SKT-LOC56 = alloca i64, align 8
  %RAX-SKT-LOC = alloca i64, align 8
  %RDI-SKT-LOC = alloca i64, align 8
  %EAX-SKT-LOC = alloca i64, align 8
  %RDX-SKT-LOC = alloca i64, align 8
  %CL-SKT-LOC = alloca i64, align 8
  %R9B-SKT-LOC = alloca i64, align 8
  %0 = inttoptr i64 %arg1 to i8*
  %memload = load i8, i8* %0, align 1
  fence seq_cst
  %1 = and i8 %memload, %memload
  %highbit = and i8 -128, %1
  %SF = icmp ne i8 %highbit, 0
  %ZF = icmp eq i8 %1, 0
  %2 = call i8 @llvm.ctpop.i8(i8 %1)
  %3 = and i8 %2, 1
  %PF = icmp eq i8 %3, 0
  %4 = zext i8 %memload to i64
  store i64 %4, i64* %R9B-SKT-LOC, align 1
  store i64 %arg1, i64* %RDI-SKT-LOC, align 1
  %CmpZF_JE = icmp eq i1 %ZF, true
  br i1 %CmpZF_JE, label %bb.8, label %bb.1

bb.1:                                        ; preds = %entry
  %5 = inttoptr i64 %arg2 to i8*
  %memload1 = load i8, i8* %5, align 1
  fence seq_cst
  %6 = and i8 %memload1, %memload1
  %highbit2 = and i8 -128, %6
  %SF3 = icmp ne i8 %highbit2, 0
  %ZF4 = icmp eq i8 %6, 0
  %7 = call i8 @llvm.ctpop.i8(i8 %6)
  %8 = and i8 %7, 1
  %PF5 = icmp eq i8 %8, 0
  %CmpZF_JE58 = icmp eq i1 %ZF4, true
  br i1 %CmpZF_JE58, label %bb.9, label %bb.2

bb.2:                                        ; preds = %bb.1
  %RSI = add i64 %arg2, 1
  %9 = call { i64, i1 } @llvm.uadd.with.overflow.i64(i64 %arg2, i64 1)
  %CF = extractvalue { i64, i1 } %9, 1
  %10 = and i64 %RSI, 255
  %11 = call i64 @llvm.ctpop.i64(i64 %10)
  %12 = and i64 %11, 1
  %PF6 = icmp eq i64 %12, 0
  %ZF7 = icmp eq i64 %RSI, 0
  %highbit8 = and i64 -9223372036854775808, %RSI
  %SF9 = icmp ne i64 %highbit8, 0
  %13 = call { i64, i1 } @llvm.sadd.with.overflow.i64(i64 %arg2, i64 1)
  %OF = extractvalue { i64, i1 } %13, 1
  %14 = zext i32 0 to i64
  store i64 %14, i64* %EAX-SKT-LOC, align 1
  %15 = zext i32 0 to i64
  store i64 %15, i64* %EAX-SKT-LOC56, align 1
  br label %bb.3

bb.3:                                        ; preds = %bb.2, %bb.6
  %ECX = zext i8 %memload1 to i32
  %16 = zext i32 %ECX to i64
  store i64 %16, i64* %CL-SKT-LOC, align 1
  store i64 %RSI, i64* %RDX-SKT-LOC, align 1
  br label %bb.4

bb.4:                                        ; preds = %bb.3, %bb.5
  %17 = load i64, i64* %R9B-SKT-LOC, align 1
  %R9B = trunc i64 %17 to i8
  %18 = load i64, i64* %CL-SKT-LOC, align 1
  %CL = trunc i64 %18 to i8
  %19 = sub i8 %R9B, %CL
  %20 = call { i8, i1 } @llvm.usub.with.overflow.i8(i8 %R9B, i8 %CL)
  %CF10 = extractvalue { i8, i1 } %20, 1
  %ZF11 = icmp eq i8 %19, 0
  %highbit12 = and i8 -128, %19
  %SF13 = icmp ne i8 %highbit12, 0
  %21 = call { i8, i1 } @llvm.ssub.with.overflow.i8(i8 %R9B, i8 %CL)
  %OF14 = extractvalue { i8, i1 } %21, 1
  %22 = call i8 @llvm.ctpop.i8(i8 %19)
  %23 = and i8 %22, 1
  %PF15 = icmp eq i8 %23, 0
  %CmpZF_JE59 = icmp eq i1 %ZF11, true
  br i1 %CmpZF_JE59, label %bb.11, label %bb.5

bb.5:                                        ; preds = %bb.4
  %RDX = load i64, i64* %RDX-SKT-LOC, align 1
  %24 = inttoptr i64 %RDX to i32*
  %memload16 = load i32, i32* %24, align 1
```

```
  fence seq_cst
  %25 = trunc i32 %memload16 to i8
  %ECX17 = zext i8 %25 to i32
  %RDX24 = add i64 %RDX, 1
  %26 = call { i64, i1 } @llvm.uadd.with.overflow.i64(i64 %RDX, i64 1)
  %CF18 = extractvalue { i64, i1 } %26, 1
  %27 = and i64 %RDX24, 255
  %28 = call i64 @llvm.ctpop.i64(i64 %27)
  %29 = and i64 %28, 1
  %PF19 = icmp eq i64 %29, 0
  %ZF20 = icmp eq i64 %RDX24, 0
  %highbit21 = and i64 -9223372036854775808, %RDX24
  %SF22 = icmp ne i64 %highbit21, 0
  %30 = call { i64, i1 } @llvm.sadd.with.overflow.i64(i64 %RDX, i64 1)
  %OF23 = extractvalue { i64, i1 } %30, 1
  %31 = trunc i32 %ECX17 to i8
  %32 = trunc i32 %ECX17 to i8
  %33 = and i8 %31, %32
  %highbit25 = and i8 -128, %33
  %SF26 = icmp ne i8 %highbit25, 0
  %ZF27 = icmp eq i8 %33, 0
  %34 = call i8 @llvm.ctpop.i8(i8 %33)
  %35 = and i8 %34, 1
  %PF28 = icmp eq i8 %35, 0
  %CmpZF_JNE = icmp eq i1 %ZF27, false
  %36 = zext i32 %ECX17 to i64
  store i64 %36, i64* %CL-SKT-LOC, align 1
  store i64 %RDX24, i64* %RDX-SKT-LOC, align 1
  br i1 %CmpZF_JNE, label %bb.4, label %bb.6

bb.6:                                            ; preds = %bb.5
  %37 = load i64, i64* %EAX-SKT-LOC, align 1
  %EAX = trunc i64 %37 to i32
  %EAX35 = add i32 %EAX, 1
  %38 = call { i32, i1 } @llvm.uadd.with.overflow.i32(i32 %EAX, i32 1)
  %CF29 = extractvalue { i32, i1 } %38, 1
  %39 = and i32 %EAX35, 255
  %40 = call i32 @llvm.ctpop.i32(i32 %39)
  %41 = and i32 %40, 1
  %PF30 = icmp eq i32 %41, 0
  %ZF31 = icmp eq i32 %EAX35, 0
  %highbit32 = and i32 -2147483648, %EAX35
  %SF33 = icmp ne i32 %highbit32, 0
  %42 = call { i32, i1 } @llvm.sadd.with.overflow.i32(i32 %EAX, i32 1)
  %OF34 = extractvalue { i32, i1 } %42, 1
  %RDI = load i64, i64* %RDI-SKT-LOC, align 1
  %memref-disp = add i64 %RDI, 1
  %43 = inttoptr i64 %memref-disp to i8*
  %memload36 = load i8, i8* %43, align 1
  fence seq_cst
  %RDI43 = add i64 %RDI, 1
  %44 = call { i64, i1 } @llvm.uadd.with.overflow.i64(i64 %RDI, i64 1)
  %CF37 = extractvalue { i64, i1 } %44, 1
  %45 = and i64 %RDI43, 255
  %46 = call i64 @llvm.ctpop.i64(i64 %45)
  %47 = and i64 %46, 1
  %PF38 = icmp eq i64 %47, 0
  %ZF39 = icmp eq i64 %RDI43, 0
  %highbit40 = and i64 -9223372036854775808, %RDI43
  %SF41 = icmp ne i64 %highbit40, 0
  %48 = call { i64, i1 } @llvm.sadd.with.overflow.i64(i64 %RDI, i64 1)
  %OF42 = extractvalue { i64, i1 } %48, 1
  %49 = and i8 %memload36, %memload36
  %highbit44 = and i8 -128, %49
  %SF45 = icmp ne i8 %highbit44, 0
  %ZF46 = icmp eq i8 %49, 0
  %50 = call i8 @llvm.ctpop.i8(i8 %49)
  %51 = and i8 %50, 1
  %PF47 = icmp eq i8 %51, 0
  %52 = zext i32 %EAX35 to i64
  store i64 %52, i64* %EAX-SKT-LOC56, align 1
  %CmpZF_JNE60 = icmp eq i1 %ZF46, false
  %53 = zext i32 %EAX35 to i64
  store i64 %53, i64* %EAX-SKT-LOC, align 1
  store i64 %RDI43, i64* %RDI-SKT-LOC, align 1
  %54 = zext i8 %memload36 to i64
  store i64 %54, i64* %R9B-SKT-LOC, align 1
  br i1 %CmpZF_JNE60, label %bb.3, label %bb.7

bb.7:                                            ; preds = %bb.6
  br label %bb.11

bb.9:                                            ; preds = %bb.1
  %55 = zext i32 0 to i64
  store i64 %55, i64* %RAX-SKT-LOC, align 1
  br label %bb.10

bb.10:                                           ; preds = %bb.9, %bb.10
  %RAX = load i64, i64* %RAX-SKT-LOC, align 1
  %memref-basereg = add i64 %arg1, %RAX
  %memref-disp48 = add i64 %memref-basereg, 1
  %56 = inttoptr i64 %memref-disp48 to i8*
```

```
%57 = load i8 , i8∗ %56, align 1
 fence seq_cst
%58 = zext i8 %57 to i64
%59 = zext i8 0 to i64
%60 = sub i64 %58, %59
%61 = call { i64 , i1 } @llvm.usub.with.overflow.i64(i64 %58, i64 %59)
%CF49 = extractvalue { i64 , i1 } %61, 1
%ZF50 = icmp eq i64 %60, 0
%highbit51 = and i64 −9223372036854775808, %60
%SF52 = icmp ne i64 %highbit51 , 0
%62 = call { i64 , i1 } @llvm.ssub.with.overflow.i64(i64 %58, i64 %59)
%OF53 = extractvalue { i64 , i1 } %62, 1
%63 = and i64 %60, 255
%64 = call i64 @llvm.ctpop.i64(i64 %63)
%65 = and i64 %64, 1
%PF54 = icmp eq i64 %65, 0
%memref−disp55 = add i64 %RAX, 1
 store i64 %memref−disp55 , i64∗ %EAX−SKT−LOC56, align 1
%CmpZF_JNE61 = icmp eq i1 %ZF50, false
 store i64 %memref−disp55 , i64∗ %RAX−SKT−LOC, align 1
 br i1 %CmpZF_JNE61, label %bb.10 , label %bb.11

bb.11:                                        ; preds = %bb.10 , %bb.7 , %bb.4
%66 = load i64 , i64∗ %EAX−SKT−LOC56, align 1
%EAX57 = trunc i64 %66 to i32
 br label %UnifiedReturnBlock

bb.8:                                         ; preds = %entry
 br label %UnifiedReturnBlock

UnifiedReturnBlock:                           ; preds = %bb.8 , %bb.11
%UnifiedRetVal = phi i32 [ %EAX57, %bb.11 ], [ 0, %bb.8 ]
 ret i32 %UnifiedRetVal
}

; Function Attrs: nofree nosync nounwind readnone speculatable willreturn
declare i8 @llvm.ctpop.i8(i8) #0

; Function Attrs: nofree nosync nounwind readnone speculatable willreturn
declare { i64 , i1 } @llvm.uadd.with.overflow.i64(i64 , i64) #0

; Function Attrs: nofree nosync nounwind readnone speculatable willreturn
declare i64 @llvm.ctpop.i64(i64) #0

; Function Attrs: nofree nosync nounwind readnone speculatable willreturn
declare { i64 , i1 } @llvm.sadd.with.overflow.i64(i64 , i64) #0

; Function Attrs: nofree nosync nounwind readnone speculatable willreturn
declare { i8 , i1 } @llvm.usub.with.overflow.i8(i8 , i8) #0

; Function Attrs: nofree nosync nounwind readnone speculatable willreturn
declare { i8 , i1 } @llvm.ssub.with.overflow.i8(i8 , i8) #0

; Function Attrs: nofree nosync nounwind readnone speculatable willreturn
declare { i32 , i1 } @llvm.uadd.with.overflow.i32(i32 , i32) #0

; Function Attrs: nofree nosync nounwind readnone speculatable willreturn
declare i32 @llvm.ctpop.i32(i32) #0

; Function Attrs: nofree nosync nounwind readnone speculatable willreturn
declare { i32 , i1 } @llvm.sadd.with.overflow.i32(i32 , i32) #0

; Function Attrs: nofree nosync nounwind readnone speculatable willreturn
declare { i64 , i1 } @llvm.usub.with.overflow.i64(i64 , i64) #0

; Function Attrs: nofree nosync nounwind readnone speculatable willreturn
declare { i64 , i1 } @llvm.ssub.with.overflow.i64(i64 , i64) #0

attributes #0 = { nofree nosync nounwind readnone speculatable willreturn }
```

Listing 14: LLVM IR predicted by GPT-4 (does not compile).

```
%%% compiler error : 3.  Running pass 'X86 DAG−>DAG Instruction Selection ' on function '@str_cspn'
 . . .
 clang : error : clang frontend command failed due to signal

declare i32 @strlen(i8∗)

define i32 @str_cspn(i8∗ %s , i8∗ %charset) {
entry :
%0 = call i32 @strlen(i8∗ %s)
%1 = call i32 @strlen(i8∗ %charset)
%2 = icmp eq i32 %0, 0
 br i1 %2, label %return_zero , label %outer_loop

outer_loop:                                   ; preds = %entry , %outer_loop
%i.0 = phi i32 [ 0, %entry ], [ %next_i , %outer_loop ]
%3 = getelementptr i8 , i8∗ %s , i32 %i.0
%4 = load i8 , i8∗ %3
%5 = icmp eq i8 %4, 0
```

```
  br i1 %5, label %return_i , label %inner_loop

inner_loop:                                      ; preds = %outer_loop , %inner_loop
  %j.0 = phi i32 [ 0, %outer_loop ] , [ %next_j , %inner_loop ]
  %6 = getelementptr i8 , i8* %charset , i32 %j.0
  %7 = load i8 , i8* %6
  %8 = icmp eq i8 %7, 0
  br i1 %8, label %inner_loop_exit , label %inner_loop_body

inner_loop_body:                                 ; preds = %inner_loop
  %9 = icmp eq i8 %4, %7
  br i1 %9, label %return_i , label %inner_loop_continue

inner_loop_continue:                             ; preds = %inner_loop_body
  %next_j = add i32 %j.0 , 1
  br label %inner_loop

inner_loop_exit:                                 ; preds = %inner_loop
  %next_i = add i32 %i.0 , 1
  br label %outer_loop

return_i:                                        ; preds = %inner_loop_body , %outer_loop
  br label %return

return_zero:                                     ; preds = %entry
  br label %return

return:                                          ; preds = %return_i , %return_zero
  %ret_val.0 = phi i32 [ %i.0 , %return_i ] , [ 0, %return_zero ]
  ret i32 %ret_val.0
}
```

### G.3  Benchmark `multiply_evens` (Forklift Fails, Lasagne Works)

In this benchmark, the function: `(arr[i] - arr2[i]) * arr2[i]` is computed for cases where `(arr[i] - arr2[i])` is even (Listing 15). This code is not vectorized by LLVM on O3 (Listing 16) due to the branch that guards the access to the out array . As a result, Lasagne is able to generate correct results for this benchmark — all the instructions are within its realm of capability (Listing 19).

However, in this case, `Forklift` generates incorrect code (Listing 18). The type signatures are equivalent, but the behavior of the function is incorrect. Again, we can see that `Forklift` produces code that relies on simple structs.

GPT-4 produces code that does not compile: it is syntactically invalid: it declares the function twice, and then produces an error when compiling one of the instructions (Listing 20).

Listing 15: Original C

```
void multiply_evens(int *arr , int *arr2 , int n, int *out) {
  int filtered = 0;
  for (int i = 0; i < n; ++i) {
        int x = arr[i] - arr2[i];
        if (x % 2 == 0) {
              out[filtered] = x * arr2[filtered];
        }
  }
}
```

Listing 16: Clang x86 O3

```
        .globl   multiply_evens          # -- Begin function multiply_evens
        .p2align         4, 0x90
        .type    multiply_evens ,@function
multiply_evens:                          # @multiply_evens
        .cfi_startproc
# %bb.0:
        testl    %edx , %edx
        jle      .LBB0_6
# %bb.1:
        movl     %edx , %r9d
        movl     %r9d , %r8d
        andl     $1, %r8d
        cmpl     $1, %edx
        jne      .LBB0_7
# %bb.2:
        xorl     %edx , %edx
.LBB0_3:
        testq    %r8, %r8
        je       .LBB0_6
# %bb.4:
```

```
        movl    (%rdi,%rdx,4), %eax
        subl    (%rsi,%rdx,4), %eax
        testb   $1, %al
        jne     .LBB0_6
# %bb.5:
        imull   (%rsi), %eax
        movl    %eax, (%rcx)
.LBB0_6:
        retq
.LBB0_7:
        subq    %r8, %r9
        xorl    %edx, %edx
        jmp     .LBB0_8
        .p2align        4, 0x90
.LBB0_12:                                # in Loop: Header=BB0_8 Depth=1
        addq    $2, %rdx
        cmpq    %rdx, %r9
        je      .LBB0_3
.LBB0_8:                                 # =>This Inner Loop Header: Depth=1
        movl    (%rdi,%rdx,4), %eax
        subl    (%rsi,%rdx,4), %eax
        testb   $1, %al
        je      .LBB0_9
# %bb.10:                                # in Loop: Header=BB0_8 Depth=1
        movl    4(%rdi,%rdx,4), %eax
        subl    4(%rsi,%rdx,4), %eax
        testb   $1, %al
        jne     .LBB0_12
        jmp     .LBB0_11
        .p2align        4, 0x90
.LBB0_9:                                 # in Loop: Header=BB0_8 Depth=1
        imull   (%rsi), %eax
        movl    %eax, (%rcx)
        movl    4(%rdi,%rdx,4), %eax
        subl    4(%rsi,%rdx,4), %eax
        testb   $1, %al
        jne     .LBB0_12
.LBB0_11:                                # in Loop: Header=BB0_8 Depth=1
        imull   (%rsi), %eax
        movl    %eax, (%rcx)
        jmp     .LBB0_12
.Lfunc_end0:
        .size   multiply_evens, .Lfunc_end0-multiply_evens
        .cfi_endproc
```

Listing 17: Ground truth LLVM IR Oz

```
define dso_local void @multiply_evens(i32* nocapture readonly %0, i32*
 nocapture readonly %1, i32 %2, i32* nocapture %3) local_unnamed_addr #0 {
  %5 = sext i32 %2 to i64
  br label %6

6:                                                ; preds = %21, %4
  %7 = phi i64 [ %22, %21 ], [ 0, %4 ]
  %8 = icmp slt i64 %7, %5
  br i1 %8, label %10, label %9

9:                                                ; preds = %6
  ret void

10:                                               ; preds = %6
  %11 = getelementptr inbounds i32, i32* %0, i64 %7
  %12 = load i32, i32* %11, align 4, !tbaa !2
  %13 = getelementptr inbounds i32, i32* %1, i64 %7
  %14 = load i32, i32* %13, align 4, !tbaa !2
  %15 = sub nsw i32 %12, %14
  %16 = and i32 %15, 1
  %17 = icmp eq i32 %16, 0
  br i1 %17, label %18, label %21

18:                                               ; preds = %10
  %19 = load i32, i32* %1, align 4, !tbaa !2
  %20 = mul nsw i32 %19, %15
  store i32 %20, i32* %3, align 4, !tbaa !2
  br label %21

21:                                               ; preds = %18, %10
  %22 = add nuw nsw i64 %7, 1
  br label %6
}
```

Listing 18: LLVM IR predicted by Forklift (IO incorrect but compiles)

```
IO example: [4, 0, 12, 12]
>>> arr = [5, 2, 8, 7]
>>> arr2 = [1, 2, 2, 3]
>>> n = 4
>>> out = [0] * 4  # Initialize 'out' with zeros to preallocate space
```

```
expected: [4, 0, 12, 12]
obtained: [6, 0, 0, 0]

%struct.struct0 = type { i32 }
define dso_local void @multiply_evens(%struct.struct0* nocapture
 readonly %0, %struct.struct0* nocapture readonly %1, i32 %2, i32* nocapture %3) local_unnamed_addr {
%5 = getelementptr inbounds %struct.struct0, %struct.struct0* %1, i64 0, i32 0
%6 = getelementptr inbounds %struct.struct0, %struct.struct0* %0, i64 0, i32 0
%7 = sext i32 %2 to i64
br label %8
8: ; preds = %21, %4
%9 = phi i64 [ %22, %21 ], [ 0, %4 ]
%10 = icmp slt i64 %9, %7
br i1 %10, label %11, label %23
11: ; preds = %8
%12 = getelementptr inbounds %struct.struct0, %struct.struct0* %0, i64 %9, i32 0
%13 = load i32, i32* %12, align 4
%14 = load i32, i32* %5, align 4
%15 = sub nsw i32 %13, %14
%16 = and i32 %15, 1
%17 = icmp eq i32 %16, 0
br i1 %17, label %18, label %21
18: ; preds = %11
%19 = load i32, i32* %5, align 4
%20 = mul nsw i32 %19, %15
store i32 %20, i32* %3, align 4
br label %21
21: ; preds = %18, %11
%22 = add nuw nsw i64 %9, 1
br label %8
23: ; preds = %8
ret void
}
```

Listing 19: LLVM IR predicted by Lasagne (IO correct but verbose)

```
define dso_local void @multiply_evens(i64 %arg1, i64 %arg2, i32 %arg3, i32* %arg4) {
entry:
  %RDX-SKT-LOC84 = alloca i64, align 8
  %EAX-SKT-LOC = alloca i32, align 4
  %RDX-SKT-LOC = alloca i64, align 8
  %0 = and i32 %arg3, %arg3
  %highbit = and i32 -2147483648, %0
  %SF = icmp ne i32 %highbit, 0
  %ZF = icmp eq i32 %0, 0
  %1 = and i32 %0, 255
  %2 = call i32 @llvm.ctpop.i32(i32 %1)
  %3 = and i32 %2, 1
  %PF = icmp eq i32 %3, 0
  %CmpZF_JLE = icmp eq i1 %ZF, true
  %CmpOF_JLE = icmp ne i1 %SF, false
  %ZFOrSF_JLE = or i1 %CmpZF_JLE, %CmpOF_JLE
  br i1 %ZFOrSF_JLE, label %bb.6, label %bb.1

bb.1:                                          ; preds = %entry
  %R8D = and i32 %arg3, 1
  %4 = and i32 %R8D, 255
  %5 = call i32 @llvm.ctpop.i32(i32 %4)
  %6 = and i32 %5, 1
  %PF1 = icmp eq i32 %6, 0
  %ZF2 = icmp eq i32 %R8D, 0
  %highbit3 = and i32 -2147483648, %R8D
  %SF4 = icmp ne i32 %highbit3, 0
  %7 = sub i32 %arg3, 1
  %8 = call { i32, i1 } @llvm.usub.with.overflow.i32(i32 %arg3, i32 1)
  %CF = extractvalue { i32, i1 } %8, 1
  %ZF5 = icmp eq i32 %7, 0
  %highbit6 = and i32 -2147483648, %7
  %SF7 = icmp ne i32 %highbit6, 0
  %9 = call { i32, i1 } @llvm.ssub.with.overflow.i32(i32 %arg3, i32 1)
  %OF = extractvalue { i32, i1 } %9, 1
  %10 = and i32 %7, 255
  %11 = call i32 @llvm.ctpop.i32(i32 %10)
  %12 = and i32 %11, 1
  %PF8 = icmp eq i32 %12, 0
  %CmpZF_JNE = icmp eq i1 %ZF5, false
  br i1 %CmpZF_JNE, label %bb.7, label %bb.2

bb.2:                                          ; preds = %bb.1
  %13 = zext i32 0 to i64
  store i64 %13, i64* %RDX-SKT-LOC84, align 1
  br label %bb.3

bb.7:                                          ; preds = %bb.1
  %14 = zext i32 %arg3 to i64
  %15 = zext i32 %R8D to i64
  %R9 = sub i64 %14, %15
  %16 = call { i64, i1 } @llvm.usub.with.overflow.i64(i64 %14, i64 %15)
  %CF9 = extractvalue { i64, i1 } %16, 1
  %ZF10 = icmp eq i64 %R9, 0
  %highbit11 = and i64 -9223372036854775808, %R9
```

```
  %SF12 = icmp ne i64 %highbit11, 0
  %17 = call { i64, i1 } @llvm.ssub.with.overflow.i64(i64 %14, i64 %15)
  %OF13 = extractvalue { i64, i1 } %17, 1
  %18 = and i64 %R9, 255
  %19 = call i64 @llvm.ctpop.i64(i64 %18)
  %20 = and i64 %19, 1
  %PF14 = icmp eq i64 %20, 0
  %21 = zext i32 0 to i64
  store i64 %21, i64* %RDX-SKT-LOC, align 1
  br label %bb.10

bb.10:                                              ; preds = %bb.9, %bb.7
  %RDX = load i64, i64* %RDX-SKT-LOC, align 1
  %memref-idxreg = mul i64 4, %RDX
  %22 = inttoptr i64 %arg1 to i8*
  %23 = getelementptr i8, i8* %22, i64 %memref-idxreg
  %24 = bitcast i8* %23 to i32*
  %memload = load i32, i32* %24, align 1
  fence seq_cst
  %memref-idxreg15 = mul i64 4, %RDX
  %25 = inttoptr i64 %arg2 to i8*
  %26 = getelementptr i8, i8* %25, i64 %memref-idxreg15
  %27 = bitcast i8* %26 to i32*
  %28 = load i32, i32* %27, align 1
  fence seq_cst
  %EAX = sub i32 %memload, %28
  %29 = call { i32, i1 } @llvm.usub.with.overflow.i32(i32 %memload, i32 %28)
  %CF17 = extractvalue { i32, i1 } %29, 1
  %ZF18 = icmp eq i32 %EAX, 0
  %highbit19 = and i32 -2147483648, %EAX
  %SF20 = icmp ne i32 %highbit19, 0
  %30 = call { i32, i1 } @llvm.ssub.with.overflow.i32(i32 %memload, i32 %28)
  %OF21 = extractvalue { i32, i1 } %30, 1
  %31 = and i32 %EAX, 255
  %32 = call i32 @llvm.ctpop.i32(i32 %31)
  %33 = and i32 %32, 1
  %PF22 = icmp eq i32 %33, 0
  %34 = trunc i32 %EAX to i8
  %35 = and i8 %34, 1
  %36 = call i8 @llvm.ctpop.i8(i8 %35)
  %37 = and i8 %36, 1
  %PF23 = icmp eq i8 %37, 0
  %ZF24 = icmp eq i8 %35, 0
  %highbit25 = and i8 -128, %35
  %SF26 = icmp ne i8 %highbit25, 0
  %CmpZF_JE = icmp eq i1 %ZF24, true
  br i1 %CmpZF_JE, label %bb.14, label %bb.11

bb.11:                                              ; preds = %bb.10
  %memref-idxreg27 = mul i64 4, %RDX
  %memref-basereg28 = add i64 %arg1, %memref-idxreg27
  %memref-disp = add i64 %memref-basereg28, 4
  %38 = inttoptr i64 %memref-disp to i32*
  %memload29 = load i32, i32* %38, align 1
  fence seq_cst
  %memref-idxreg30 = mul i64 4, %RDX
  %memref-basereg31 = add i64 %arg2, %memref-idxreg30
  %memref-disp32 = add i64 %memref-basereg31, 4
  %39 = inttoptr i64 %memref-disp32 to i32*
  %40 = load i32, i32* %39, align 1
  fence seq_cst
  %EAX33 = sub i32 %memload29, %40
  %41 = call { i32, i1 } @llvm.usub.with.overflow.i32(i32 %memload29, i32 %40)
  %CF34 = extractvalue { i32, i1 } %41, 1
  %ZF35 = icmp eq i32 %EAX33, 0
  %highbit36 = and i32 -2147483648, %EAX33
  %SF37 = icmp ne i32 %highbit36, 0
  %42 = call { i32, i1 } @llvm.ssub.with.overflow.i32(i32 %memload29, i32 %40)
  %OF38 = extractvalue { i32, i1 } %42, 1
  %43 = and i32 %EAX33, 255
  %44 = call i32 @llvm.ctpop.i32(i32 %43)
  %45 = and i32 %44, 1
  %PF39 = icmp eq i32 %45, 0
  %46 = trunc i32 %EAX33 to i8
  %47 = and i8 %46, 1
  %48 = call i8 @llvm.ctpop.i8(i8 %47)
  %49 = and i8 %48, 1
  %PF40 = icmp eq i8 %49, 0
  %ZF41 = icmp eq i8 %47, 0
  %highbit42 = and i8 -128, %47
  %SF43 = icmp ne i8 %highbit42, 0
  store i32 %EAX33, i32* %EAX-SKT-LOC, align 1
  %CmpZF_JNE1 = icmp eq i1 %ZF41, false
  br i1 %CmpZF_JNE1, label %bb.9, label %bb.12

bb.12:                                              ; preds = %bb.11
  br label %bb.15

bb.14:                                              ; preds = %bb.10
  %50 = inttoptr i64 %arg2 to i32*
  %memload44 = load i32, i32* %50, align 1
  fence seq_cst
```

```
%EAX45 = mul i32 %EAX, %memload44
 fence seq_cst
 store i32 %EAX45, i32* %arg4, align 1
%memref-idxreg46 = mul i64 4, %RDX
%memref-basereg47 = add i64 %arg1, %memref-idxreg46
%memref-disp48 = add i64 %memref-basereg47, 4
 %51 = inttoptr i64 %memref-disp48 to i32*
%memload49 = load i32, i32* %51, align 1
 fence seq_cst
%memref-idxreg50 = mul i64 4, %RDX
%memref-basereg51 = add i64 %arg2, %memref-idxreg50
%memref-disp52 = add i64 %memref-basereg51, 4
 %52 = inttoptr i64 %memref-disp52 to i32*
 %53 = load i32, i32* %52, align 1
 fence seq_cst
%EAX53 = sub i32 %memload49, %53
 %54 = call { i32, i1 } @llvm.usub.with.overflow.i32(i32 %memload49, i32 %53)
%CF54 = extractvalue { i32, i1 } %54, 1
%ZF55 = icmp eq i32 %EAX53, 0
%highbit56 = and i32 -2147483648, %EAX53
%SF57 = icmp ne i32 %highbit56, 0
 %55 = call { i32, i1 } @llvm.ssub.with.overflow.i32(i32 %memload49, i32 %53)
%OF58 = extractvalue { i32, i1 } %55, 1
 %56 = and i32 %EAX53, 255
 %57 = call i32 @llvm.ctpop.i32(i32 %56)
 %58 = and i32 %57, 1
%PF59 = icmp eq i32 %58, 0
 %59 = trunc i32 %EAX53 to i8
 %60 = and i8 %59, 1
 %61 = call i8 @llvm.ctpop.i8(i8 %60)
 %62 = and i8 %61, 1
%PF60 = icmp eq i8 %62, 0
%ZF61 = icmp eq i8 %60, 0
%highbit62 = and i8 -128, %60
%SF63 = icmp ne i8 %highbit62, 0
 store i32 %EAX53, i32* %EAX-SKT-LOC, align 1
%CmpZF_JNE2 = icmp eq i1 %ZF61, false
 br i1 %CmpZF_JNE2, label %bb.9, label %bb.15

bb.15:                                          ; preds = %bb.14, %bb.12
%EAX64 = load i32, i32* %EAX-SKT-LOC, align 1
 %63 = inttoptr i64 %arg2 to i32*
%memload65 = load i32, i32* %63, align 1
 fence seq_cst
%EAX66 = mul i32 %EAX64, %memload65
 fence seq_cst
 store i32 %EAX66, i32* %arg4, align 1
 br label %bb.9

bb.9:                                           ; preds = %bb.15, %bb.14, %bb.11
%RDX73 = add i64 %RDX, 2
 %64 = call { i64, i1 } @llvm.uadd.with.overflow.i64(i64 %RDX, i64 2)
%CF67 = extractvalue { i64, i1 } %64, 1
 %65 = and i64 %RDX73, 255
 %66 = call i64 @llvm.ctpop.i64(i64 %65)
 %67 = and i64 %66, 1
%PF68 = icmp eq i64 %67, 0
%ZF69 = icmp eq i64 %RDX73, 0
%highbit70 = and i64 -9223372036854775808, %RDX73
%SF71 = icmp ne i64 %highbit70, 0
 %68 = call { i64, i1 } @llvm.sadd.with.overflow.i64(i64 %RDX, i64 2)
%OF72 = extractvalue { i64, i1 } %68, 1
 %69 = sub i64 %R9, %RDX73
 %70 = call { i64, i1 } @llvm.usub.with.overflow.i64(i64 %R9, i64 %RDX73)
%CF74 = extractvalue { i64, i1 } %70, 1
%ZF75 = icmp eq i64 %69, 0
%highbit76 = and i64 -9223372036854775808, %69
%SF77 = icmp ne i64 %highbit76, 0
 %71 = call { i64, i1 } @llvm.ssub.with.overflow.i64(i64 %R9, i64 %RDX73)
%OF78 = extractvalue { i64, i1 } %71, 1
 %72 = and i64 %69, 255
 %73 = call i64 @llvm.ctpop.i64(i64 %72)
 %74 = and i64 %73, 1
%PF79 = icmp eq i64 %74, 0
 store i64 %RDX73, i64* %RDX-SKT-LOC84, align 1
%CmpZF_JE3 = icmp eq i1 %ZF75, true
 store i64 %RDX73, i64* %RDX-SKT-LOC, align 1
 br i1 %CmpZF_JE3, label %bb.3, label %bb.10

bb.3:                                           ; preds = %bb.2, %bb.9
 %75 = zext i32 %R8D to i64
 %76 = zext i32 %R8D to i64
 %77 = and i64 %75, %76
%highbit80 = and i64 -9223372036854775808, %77
%SF81 = icmp ne i64 %highbit80, 0
%ZF82 = icmp eq i64 %77, 0
 %78 = and i64 %77, 255
 %79 = call i64 @llvm.ctpop.i64(i64 %78)
 %80 = and i64 %79, 1
%PF83 = icmp eq i64 %80, 0
%CmpZF_JE4 = icmp eq i1 %ZF82, true
 br i1 %CmpZF_JE4, label %bb.6, label %bb.4
```

```
bb.4:                                                         ; preds = %bb.3
  %RDX85 = load i64 , i64∗ %RDX−SKT−LOC84 , align 1
  %memref−idxreg86 = mul i64 4 , %RDX85
  %81 = inttoptr i64 %arg1 to i8∗
  %82 = getelementptr i8 , i8∗ %81 , i64 %memref−idxreg86
  %83 = bitcast i8∗ %82 to i32∗
  %memload88 = load i32 , i32∗ %83 , align 1
  fence seq_cst
  %memref−idxreg89 = mul i64 4 , %RDX85
  %84 = inttoptr i64 %arg2 to i8∗
  %85 = getelementptr i8 , i8∗ %84 , i64 %memref−idxreg89
  %86 = bitcast i8∗ %85 to i32∗
  %87 = load i32 , i32∗ %86 , align 1
  fence seq_cst
  %EAX91 = sub i32 %memload88 , %87
  %88 = call { i32 , i1 } @llvm.usub.with.overflow.i32(i32 %memload88 , i32 %87)
  %CF92 = extractvalue { i32 , i1 } %88 , 1
  %ZF93 = icmp eq i32 %EAX91 , 0
  %highbit94 = and i32 −2147483648 , %EAX91
  %SF95 = icmp ne i32 %highbit94 , 0
  %89 = call { i32 , i1 } @llvm.ssub.with.overflow.i32(i32 %memload88 , i32 %87)
  %OF96 = extractvalue { i32 , i1 } %89 , 1
  %90 = and i32 %EAX91 , 255
  %91 = call i32 @llvm.ctpop.i32(i32 %90)
  %92 = and i32 %91 , 1
  %PF97 = icmp eq i32 %92 , 0
  %93 = trunc i32 %EAX91 to i8
  %94 = and i8 %93 , 1
  %95 = call i8 @llvm.ctpop.i8(i8 %94)
  %96 = and i8 %95 , 1
  %PF98 = icmp eq i8 %96 , 0
  %ZF99 = icmp eq i8 %94 , 0
  %highbit100 = and i8 −128 , %94
  %SF101 = icmp ne i8 %highbit100 , 0
  %CmpZF_JNE5 = icmp eq i1 %ZF99 , false
  br i1 %CmpZF_JNE5 , label %bb.6 , label %bb.5

bb.5:                                                         ; preds = %bb.4
  %97 = inttoptr i64 %arg2 to i32∗
  %memload102 = load i32 , i32∗ %97 , align 1
  fence seq_cst
  %EAX103 = mul i32 %EAX91 , %memload102
  fence seq_cst
  store i32 %EAX103 , i32∗ %arg4 , align 1
  br label %bb.6

bb.6:                                                         ; preds = %bb.5 , %bb.4 , %bb.3 , %entry
  ret void
}

; Function Attrs : nofree nosync nounwind readnone speculatable willreturn
declare i32 @llvm.ctpop.i32(i32) #0

; Function Attrs : nofree nosync nounwind readnone speculatable willreturn
declare { i32 , i1 } @llvm.usub.with.overflow.i32(i32 , i32) #0

; Function Attrs : nofree nosync nounwind readnone speculatable willreturn
declare { i32 , i1 } @llvm.ssub.with.overflow.i32(i32 , i32) #0

; Function Attrs : nofree nosync nounwind readnone speculatable willreturn
declare { i64 , i1 } @llvm.usub.with.overflow.i64(i64 , i64) #0

; Function Attrs : nofree nosync nounwind readnone speculatable willreturn
declare { i64 , i1 } @llvm.ssub.with.overflow.i64(i64 , i64) #0

; Function Attrs : nofree nosync nounwind readnone speculatable willreturn
declare i64 @llvm.ctpop.i64(i64) #0

; Function Attrs : nofree nosync nounwind readnone speculatable willreturn
declare i8 @llvm.ctpop.i8(i8) #0

; Function Attrs : nofree nosync nounwind readnone speculatable willreturn
declare { i64 , i1 } @llvm.uadd.with.overflow.i64(i64 , i64) #0

; Function Attrs : nofree nosync nounwind readnone speculatable willreturn
declare { i64 , i1 } @llvm.sadd.with.overflow.i64(i64 , i64) #0

attributes #0 = { nofree nosync nounwind readnone speculatable willreturn }
```

Listing 20: LLVM IR predicted by GPT-4 (does not compile).

```
. Aside from ilegally redeclaring the function , the function definition itself doesnt compile)
oai_m_evens.ll:13:1: error: label expected to be numbered '8'
10:                                                           ; preds = %12, %5
^

declare void @multiply_evens(i32∗, i32∗, i32∗, i64)

define void @multiply_evens(i32∗ %0, i32∗ %1, i32∗ %2, i64 %3) {
```

```
start:
  %4 = icmp sle i64 %3, 0
  br i1 %4, label %14, label %5

5:                                                  ; preds = %start
  %6 = add i64 %3, −1
  %7 = xor i64 %3, %3
  br label %10

10:                                                 ; preds = %12, %5
  %8 = phi i64 [ %7, %5 ], [ %13, %12 ]
  %9 = getelementptr i32 , i32∗ %0, i64 %8
  %10 = load i32 , i32∗ %9
  %11 = getelementptr i32 , i32∗ %1, i64 %8
  %12 = load i32 , i32∗ %11
  %13 = sub i32 %10, %12
  %14 = and i32 %13, 1
  %15 = icmp eq i32 %14, 0
  br i1 %15, label %16, label %12

16:                                                 ; preds = %10
  %17 = load i32 , i32∗ %1
  %18 = mul i32 %17, %13
  %19 = getelementptr i32 , i32∗ %2, i64 %8
  store i32 %18, i32∗ %19
  br label %12

12:                                                 ; preds = %16, %10
  %13 = add i64 %8, 1
  %20 = icmp eq i64 %13, %6
  br i1 %20, label %21, label %10

21:                                                 ; preds = %12
  ret void
}
```

