# OpenReview forum: "Forklift: An Extensible Neural Lifter"
_colmweb.org/COLM/2024/Conference — COLM_

### Official Review · Reviewer_GXRk · 2024-05-10

**Rating:** 7
**Confidence:** 4
**Ethics Flag:** 1

**Summary:**

This paper presents Forklift, a model designed to translate machine architecture-dependent assembly code into an intermediate representation that is architecture-independent, facilitating the migration of (legacy) programs to other machine architectures. Forklift is based on an encoder-decoder Transformer architecture and is trained using a parallel dataset created by the authors. The authors also incorporate the idea of incremental learning, initially training the model with x86 architecture and then fine-tuning it with another architecture. Experimental results demonstrate that Forklift outperforms other baselines, including a state-of-the-art hand-written model and GPT4, particularly in terms of passing unit tests. Furthermore, the authors present results from additional experiments to address interesting questions, such as the impact of incremental learning and error analysis.

**Questions To Authors:**

- The incremental learning idea is intriguing. Would training the model at the optimization level be feasible? For instance, the paper suggests the order: x86 => ARM => RISC-V. How about this alternative order: x86 O0 => x86 O3 => ARM O0 => ARM O3? Since the optimization level varies even with the same architecture, I'm curious about the potential results.

- In my experience, tokenization poses a significant challenge in improving the performance of understanding assembly code. How does the model's performance differ when using Byte Pair Encoding (BPE) compared to a unigram tokenizer?

- What are the features listed in Table 8? Could you provide more information about "edit_sim" and "has_float"?

**Reasons To Accept:**

- The paper introduces an interesting problem: mitigating legacy software formed by architecture-dependent assembly code, which could inspire further research by other researchers.

- The paper presents a simple and effective method for solving the problem, providing a solid foundation for future research extensions.

- The paper demonstrates various experimental results supporting the usefulness of Forklift.

**Reasons To Reject:**

- Lack of detailed description of the parallel datasets and evaluation datasets, particularly unit tests. The number of data items and the difficulty level of unit tests are not specified, and it's unclear whether the authors have made the datasets open for reproducibility.

- Some questions I have for the authors:

---

> ### Author Rebuttal · Authors · 2024-05-31
>
> Thank you for your feedback.
>
> ### Lack of parallel dataset details
>
> We use the C programs and unit tests, as described in Exebench [1]. This is a publicly available open-source benchmark suite with about 4M compilable functions that we augment with additional assembly and LLVM IR (which we will release). The corresponding unit tests are also public. The input/output pairs for the Synth dataset are also public. In both datasets, there are 10 unit tests per function. In both cases, not only return values are tested, but also arrays modified in place. Additionally, for Exebench, global variables are also checked for potential mutations (Synth has no global variables).
>
> We compile each program to 3 different ISAs (x86, ARM, and RISC-V)  with two compilers (GCC and LLVM)  and three optimization levels (Oz, O0, and O3).  We will make all data public and clarify. We will  also provide more statistics about the data in the paper itself.
>
> ### Incremental learning order.
>
>
>
> We experimented with x86 O3 => ARM O3 => RISC-V O3. The motivation for this order is that  x86 is the established/legacy architecture with ARM and then  RISC-V emerging later. We did not consider O0 as in reality, vendor released  binaries are  optimized. Furthermore, optimized binaries  have been shown  to be  more challenging to translate. We would expect to see similar results with learning strategies involving O0. Instead, we included compiler transfer results (Clang -> GCC) in section 5.3.
>
> ### Tokenizer
>
> We used Unigram rather than BPE as [2] showed it to perform marginally better. Additionally, we added custom regex split rules for the pre-tokenizer to avoid inconsistent digit/register tokenization and remove boilerplate characters from assembly/LLVM IR.  BPE is unlikely to make a significant difference. We will add a discussion to the paper.
>
> ### Features
>
> Edit_sim means edit similarity, which is defined as 1- edit distance. Has_float means the code contains floating point operations. We will provide a full description of Table 8 entries in the paper, originally limited by the length limit.
>
> [1] Armengol-Estape et al. ExeBench: an ML-scale dataset of executable C functions. https://dl.acm.org/doi/abs/10.1145/3520312.3534867
>
> [2] Bostrom et al. Byte Pair Encoding is Suboptimal for Language Model Pretraining. https://aclanthology.org/2020.findings-emnlp.414/

---

> > ### Comment · Reviewer_GXRk · 2024-06-03
> >
> > Thank you for addressing my questions. I will improve my score as the authors have clarified my concerns. I hope they will incorporate the comments from the other reviewers.
> >
> > Yours,

---

### Official Review · Reviewer_9Een · 2024-05-11

**Rating:** 7
**Confidence:** 4
**Ethics Flag:** 1

**Summary:**

The paper introduces Forklift, which converts assembly into an intermediate representation for generic compiler optimization across different hardware platforms. Forklift outperforms Lasagne and GPT-4, gradually adding support for new ISAs by fine-tuning the assembly encoder and freezing the IR decoder to improve overall accuracy and efficiency. Overall, it leans towards an application-oriented approach.

**Reasons To Accept:**

1. The addressed problem is of practical significance and has the potential to become part of the software's foundational infrastructure.
2. The experimental results are impressive, surpassing previous methods and the latest large language models, demonstrating the effectiveness of Forklift.

**Reasons To Reject:**

1. There are certain limitations in the methodological contribution. Could some structured learning modules be introduced based on the representation of the intermediate language? Will the increase in data volume continue to improve effectiveness, i.e., is there a scaling law in the size of intermediate representation data?
2. Experiments were not conducted on popular code LLMs. For instance, StarCoder2 was pre-trained on LLVM. Would this be beneficial for the task?

---

> ### Author Rebuttal · Authors · 2024-05-31
>
> Thank you for your feedback.
>
> ### Structured Learning Modules
>
> While the idea of structured modules such as GNNs for code is interesting, following previous work (e.g. [2]) we relied on a token/sequence-based approach with Transformers, as it’s currently the state of the art for generative code settings.  However, it is possible that there are structured models that are more effective than text-based models for assembly and is an open question.  We will add a discussion.
>
> ### Scaling Laws for Size of Intermediate Representation
>
> There are two dimensions that we can scale the data size in: the size of the individual examples and the number of training instances.
>
> For the size of individual examples, in Table 8 we provide the correlation between the length of the original C code (correlated with both the length of assembly and LLVM IR) and the IO accuracy, showing that Forklift has a negative accuracy correlation with length. Interestingly, we observed that this was not only due to the context window limitation: longer examples are more difficult to handle for the model, even if they fit in the context window. GPT-4, for which all examples fit in their context window, also has a negative accuracy correlation with length. We will clarify and add data on the relation between performance and length.
>
> For the number of samples used in training, Figure 2 in the Appendix shows Input/output accuracy on different stages in the training on a subset of ExeBench for ARM (Figure 3 shows the same for Synth). Again we will add clarification.
>
> ### Other LLMs
>
>
> We followed the findings in [1], in which an encoder-decoder trained from scratch clearly outperformed (e.g., 68.9% compared to 11.1%/8.9% or 81.2% compared to 36.4%) strong fine-tuned code LLMs baselines (CodeLlama, Starcoder) in a binary translation task. Recent  code LLMs are starting to consider the inclusion of low-level languages in pretraining datasets, previously typically neglected,. Starcoder2, a concurrent work to ours, is a step in the right direction, but the LLVM IR is not paired with the corresponding assembly, and the proportion of runtime-optimized assembly is tiny. We will add a discussion to the final paper.
>
>
> [1] Lee et al. Guess & Sketch: Language Model Guided Transpilation. https://openreview.net/forum?id=qPFsIbF3V6
> [2] Lachaux et al. https://proceedings.neurips.cc/paper/2020/file/ed23fbf18c2cd35f8c7f8de44f85c08d-Paper.pdf

---

### Official Review · Reviewer_WGiA · 2024-05-13

**Rating:** 7
**Confidence:** 3
**Ethics Flag:** 1

**Summary:**

The paper proposes Forklift, a neural lifter using a Transformer model to translate assembly code of different ISAs to the architecture-independent LLVM IR. Forklift achieves better accuracy and efficiency than both a state-of-the-art hand-written lifter and GPT-4 when translating x86 programs and enables translation from additional ISAs.

**Questions To Authors:**

- How do you ensure that there is no leakage between the training data and the evaluation data?
- It’s great that Forklift performs significantly better than Lasagne out-of-the-box. Do you have a sense of whether the failures of Lasagne that you identified in this could be fixed? Do you know what it would take and how much better the performance would be?

**Reasons To Accept:**

- This paper demonstrates the viability of lifting assembly code to LLVM IR using neural language models. The discussion on which IR language and IR optimization level works best is valuable.
- The training data collected by this paper would be useful to the community working on neural code generation.
- The discussion on fine-tuning the assembly encoder while keeping the IR decoder fixed is interesting and seems to provide some benefits in terms of accuracy and efficiency.

**Reasons To Reject:**

- The main novelty in this paper is the application along with its setup (training and evaluation data, etc.) while there is little to no innovation over standard LM training.

---

> ### Author Rebuttal · Authors · 2024-05-31
>
> Thank you for your feedback.
>
> ### Novelty
>
> The main novelty of this paper is the demonstration that purely neural approaches (Transformers) can address a core challenge in binary portability: replacing the time-consuming task of manually writing lifting rules with an automatically learned approach targeting an intermediate language rather than another target directly, so that compiler infrastructure can be reused.
>
> To tackle this challenge, we introduce several novel features.  We leverage incremental learning, where models that are initially trained for one assembler can be retrained to run on a new assembler, reusing the LLVM IR decoder component.  We demonstrate that for assembly, there is significant benefit in taking this incremental learning approach between architectures.  We also augmented the Exebench dataset [1] with LLVM IR and architecture targets and optimization levels not previously present in the dataset. Forklift is the first model that is capable of lifting from assembly.  LLVM IR is also a challenging target for language models, as it lacks the structure and readability that language models for code are typically able to take advantage of. We will update the paper to highlight novelty.
>
> ### Data leakage
>
> The data set in [1] is claimed to be constructed such that there is no overlap, based on exact token-level deduplication with some normalizations (e.g. space normalization, comment removal), and a similar strategy was applied for Synth. We run an additional deduplication not done in the original dataset: assembly-level deduplication (rather than C-level deduplication, as different C functions can be lowered to the same assembly). We will clarify this in the paper.
>
> ### Lasagne failures
>
> Generally, Lasagne failures are not due to specific bugs that could be fixed, but due to unsupported features that could be implemented but would require significant engineering effort. Lasagne has little to no support for vectorized instructions, and there are large bodies of instructions that are not supported in its ARM port. Additionally, Lasagne also has issues in code involving global variables, which would require  significant new analysis to resolve. which has traditionally been a challenging source language for large language models to LLVM IR.
> We will provide a greater analysis in the final paper.
>
> [1] Armengol-Estape et al. ExeBench: an ML-scale dataset of executable C functions. https://dl.acm.org/doi/abs/10.1145/3520312.3534867

---

> > ### Comment · Reviewer_WGiA · 2024-06-02
> >
> > Thank you for the response! I think this helps clarify some of my questions.

---

### Official Review · Reviewer_zjVU · 2024-05-24

**Rating:** 7
**Confidence:** 4
**Ethics Flag:** 1

**Summary:**

This paper proposes a neural lifter to translate the source binary language into a shared compiler intermediate representation. The approach is to generate LLVM IR and source binaries from a large set of C functions, and reverse the translation from binaries to the LLVM IR.

**Questions To Authors:**

Please see and respond to reasons to reject as well as

3. Why do you think the model is experiencing negative correlation on length of original C code (c_length), and number of arguments of the function?

4. "We assume that the desired functions have reference unit tests" - I don't see any particular method here. This seems like standard approach to function execution accuracy? Also, where do you construct the unit tests? Later you write that "We then take the output LLVM IR, compile it back, and call the function from a C interface with the original context of the function. We then check for input/output accuracy" (presumably against unit tests).

**Reasons To Accept:**

1. The resulting model performs well compared to two strong baselines (manual lifter and GPT4), and if open-sourced, will be a significant contribution to not only the research community, but also industry.
2. The paper is very clearly written and easy to follow.
3. The analysis experiments are reasonable.

**Reasons To Reject:**

1. The details on 3.1 Data Generation, however are quite lacking. Can you quantify the amount of the publicly available repositories crawled and describe the type of sources? Were there any challenges in applying the method of Armengol-Estape et al., 2022?

2. I am not fully convinced by "We do not fine tune from an existing LLM due to the
additional computational cost not being justified given the poor performance of code
LLMs on low-level code observed" - I think an experiment is required is to confirm that fine tune from an existing LLM (even if poor) is worse than train from scratch.

3. The results in Table 1 look v good - was there any check for train-test overlap since you crawled from public repos?

---

> ### Author Rebuttal · Authors · 2024-05-31
>
> Thank you for your feedback.
>
> ### Data generation
>
> We took the dataset in [1] and added the corresponding versions of  LLVM IR (O0, Oz, O3) plus (x86, ARM, RISC) assembler when compiled by (GCC, Clang) with (Oz, O0, O3). The original dataset features over 4M compilable functions crawled from GitHub. The main challenge was engineering: systematically generating the data by compiling millions of functions and processing the corresponding assembly. We will clarify this in the paper.
>
> ### Fine-tuning
>
> Fine-tuning an existing model rather than training from scratch is interesting, as there is probably an overlap between binary lifting and other code-comprehension tasks. However, based on the findings in [2], in which an encoder-decoder trained from scratch clearly outperformed (e.g., 68.9% compared to 11.1%/ 8.9% or 81.2% compared to 36.4%) strong fine-tuned code LLM baselines (CodeLlama, Starcoder) in a related binary translation task
> we trained from scratch. We will clarify this in the paper.
>
> ### Results and overlap
>
> The data set in [1] is claimed to be constructed such that there is no overlap, based on exact token-level deduplication with some normalizations (e.g. space normalization, comment removal). We did an additional deduplication not done in the original dataset: assembly-level deduplication (rather than C-level deduplication, as different C functions can be lowered to the same assembly).  We will clarify these deduplication steps taken in the paper, and all the data will be public.
>
> ### Negative correlation
>
> Generally, the longer the original C, the longer the corresponding LLVM IR and assembly, and sequence modeling models typically struggle with longer sequences. Increasing the number of arguments increases the complexity of e.g. stack management code, leading to a greater risk of failure. We will clarify.
>
> ### Unit tests
>
> Unit testing is a common way of evaluating code correctness. Both evaluation sets (Exebench test set and Synth) provide unit tests. We compare the behavior of running Forklift’s translation against these. This includes not only return values but also arrays modified in place and mutated global variables. We will clarify this in the paper.
>
> [1] Armengol-Estape et al. ExeBench: an ML-scale dataset of executable C functions. https://dl.acm.org/doi/abs/10.1145/3520312.3534867
>
> [2] Lee et al. Guess & Sketch: Language Model Guided Transpilation. https://openreview.net/forum?id=qPFsIbF3V6

---

### Decision · Program_Chairs · 2024-07-10

**Decision:**

Accept

**Comment:**

This paper presents Forklift for translating assembly to LLVM IR. They discussed their approach for data generation, model architecture, and other design choices. The incremental learning method is interesting: when adding new assembly languages, they can freeze the decoder and finetune for new languages. The neural architecture is an encoder-decoder Transformer based on BART. They demonstrate strong empirical results, where their model outperforms Lasagne and GPT-4.

The data and method from this work can be great contributions to the field of LM for software engineering, and thus I recommend acceptance. I recommend the authors to revise their work according to suggestions from reviewers. In particular, I think it is more convincing to add finetuning experiments with more recent coding language models, and it is helpful to show the empirical results on how the base model and pretraining affect the performance.